

# The measurement of mean wind, variances and covariances from an instrumented mobile car in a rural environment

Stefan J. Miller[1], Mark Gordon[1]

[1]Department of Earth and Space Science and Engineering, York University, Toronto ON, M3J 1P3, Canada

*Correspondence to*: Stefan J. Miller (sjmiller@yorku.ca)

**Abstract.** On 20 and 22 Aug 2019, a small tripod was outfitted with a sonic anemometer and placed in a highway shoulder to compare with measurements made on an instrumented car as it travelled past the tripod. The rural measurement site in this investigation was selected so that the instrumented car travelled past many upwind surface obstructions and experienced the occasional passing vehicle. To obtain an accurate mean wind speed and mean wind direction on a moving car, it is necessary

to correct for flow distortion and remove the vehicle speed from the measured velocity component parallel to vehicle motion (for straight-line motion). In this study, the velocity variances and turbulent fluxes measured by the car are calculated using two approaches: (1) eddy-covariance and (2) wavelet analysis. The results show that wavelet analysis can better resolve low frequency contributions, and this leads to a reduction in the horizontal velocity variances measured on the car, giving a better estimate for some measurement averages when compared to the tripod. A wavelet–based approach to remove the effects of

sporadic passing traffic is developed and applied to a measurement period during which a heavy–duty truck passes in the opposite highway lane; removing the times with traffic in this measurement period gives approximately a 10% reduction in the turbulent kinetic energy. The vertical velocity variance and vertical turbulent heat flux measured on the car are biased low compared to the tripod. This low bias may be related to a mismatch in the flux footprint of the car versus the tripod, or perhaps related to rapid flow distortion at the measurement location on the car. When random measurement uncertainty is considered,

the vertical momentum flux is found to be consistent with the tripod in the 95% confidence interval, and significantly different than zero for most measurement periods.

## 1 Introduction

Measurements of atmospheric means, variances and covariances obtained from an instrumented mobile car can provide low cost, in situ observations close to the ground, and over a large measurement domain. Hereafter 'instrumented mobile car' refers

to all potential on–road vehicles that could serve as a measurement platform, including cars, sport utility vehicles, pickup trucks, minivans, or larger mobile laboratories that use a heavy–duty truck. Previous investigations have largely used instrumented mobile cars for the measurement of near–surface atmospheric means, but minimal attention has been given to their use for the measurement of turbulence (i.e., variances and covariances). In the nocturnal boundary layer characterized by stable conditions and weak flow, turbulence near the surface mainly originates from poorly understood non–stationary



mechanical shear and submesoscale motions (Mahrt et al., 2012; Van De Wiel et al., 2012) such as low–level jets, thermotopographic wind systems (i.e., katabatic flow) and breaking gravity waves (Salmond and McKendry, 2005). In the very–stable boundary layer the generated turbulence is often intermittent and results in the vertical transport of scalars (i.e., heat, pollutants), but stationary towers may be too isolated and "site–specific" to adequately sample the temporally and spatially localized turbulence (Salmond and McKendry, 2005). The mobile car, however, can measure along a driven path,

which may provide a more representative sample of turbulence near the surface compared to a stationary tower. In addition, the mobile car may also be used to obtain in situ wind and turbulence measurements near the surface within the urban boundary layer, measurements that may help validate high–resolution, street–level models. In the near–surface urban boundary layer, the strength of the wind and the intensity of turbulence are influenced by the composition of buildings and trees (Mochida et al., 2008; Gromke and Blocken 2015; Hertwig et al., 2019; Krayenhoff et al., 2020), and can have a significant impact on

pedestrian comfort (Hunt et al., 1976; Yu et al., 2021), and neighborhood–level pollutant dispersion (Aristodemou et al., 2018; Su et al., 2019). The mobile car involves less logistical limitations (i.e., permits, vandalism) and potentially affords a greater spatial coverage when compared to the installation of a stationary tower in a high–density urban area. Furthermore, as the resolution of numerical weather prediction models continues to improve, the measurement of localized variations in near–surface heat, momentum and moisture fluxes may improve the prediction of convective storms (Markowski et al., 2017).

The instrumented mobile car has been used in various investigations to measure atmospheric means near the surface (Bogren and Gustavsson 1991; Straka et al., 1996; Achberger and Bärring 1999; Armi and Mayr 2007; Mayr and Armi 2008; Taylor et al., 2011; Smith et al., 2009; White et al., 2014, Currey et al., 2016; de Boer et al., 2021). Gordon et al. (2012) and Miller et al. (2019) used the instrumented car for the measurement of velocity variances on highways to quantify vehicle–induced turbulence. Despite the increasing number of investigations using instrumented mobile car systems for atmospheric

measurements, there are limited studies that examine their performance and accuracy for the measurement of the mean flow, velocity variances and covariances.

Achberger and Bärring (1999) investigated the accuracy of mean temperature measurements made on a minibus in low–speed driving conditions (8 to 11 m s$^{-1}$) by installing four thermocouples at various heights (0.5 m, 1 m, 2 m, and 4 m). From their results they developed a spectral correction for the measured air temperature to remove the effects due to thermal

inertia of the thermocouples. More recently, Anderson et al. (2012) evaluated the feasibility of using passenger vehicles (9 in total) to collect mean air temperature and air pressure measurements on roads, with the end goal of improving road weather forecasts to reduce weather–related traffic fatalities. They found good agreement for mean air temperature measurements made on passenger vehicles when compared to mean air temperature measurements made by stationary weather stations, and poor agreement for air pressure.

Belušic et al. (2014) is the first known study to evaluate a three–dimensional sonic anemometer (model CSAT3, sampling frequency of 20 Hz) affixed to a passenger vehicle for its accuracy at measuring atmospheric variances and covariances, in addition to atmospheric means. In their setup the sonic anemometer was supported by a sophisticated arm and lattice aluminum frame; the arm held the sonic above the vehicle's top at a height of 3 m from the ground, positioned slightly



ahead of the vehicle's front end. Recently, Hanlon and Risk (2020) investigated how the placement of a sonic anemometer on

the vehicle affects the accuracy of velocity measurements, by applying computational fluid dynamics modelling in combination

with mobile car measurements. The anemometers were placed vertically upward on top of the vehicle's roof.

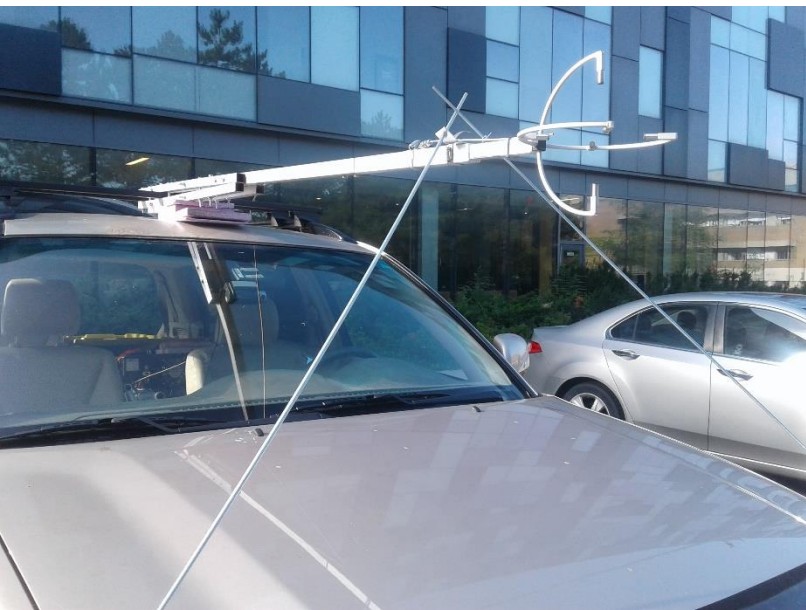

**Figure 1: A front view of instrumented car (also referred to as a mobile car platform or mobile car laboratory) used in this**
**investigation.**

If 1 min averages are assumed, then measurements (i.e., wind velocity, gas concentration) obtained from an

instrumented car travelling at near–highway speeds (i.e., 15 to 25 m s$^{-1}$) are made over a significant spatial path on the order

of $10^3$ meters, where surface variations (i.e., vegetation, building structures, other traffic) can be significant. A single spatial

path measured by the vehicle may therefore feature flow conditions that are not stationary and an upwind surface that is not

homogenous. This calls into question the applicability of the eddy–covariance (EC) method which requires near–stationary

conditions to reduce uncertainties in the estimation of variances and covariances. During their investigation Belušic et al.

(2014) made car measurements on a nearly flat, homogenous portion of remote rural highway, without traffic and without

large upwind obstacles such as trees and houses. Therefore, their investigation represented an "idealized" case. Even so, they

found instances where the car–measured horizontal velocity variances were significantly overestimated compared to

measurements made by a nearby stationary tower. They concluded that non–stationarity of the flow was the likely cause

leading to the anomalously large car–measured horizontal velocity variances. Their results demonstrate that non–stationarity

of the flow cannot be ignored when measuring on an instrumented mobile car. Recently Schaller et al. (2017) applied wavelet

analysis as an alternative technique to estimate turbulent methane fluxes measured by a fixed tower in non–stationary

conditions. For periods fulfilling the stationarity requirement, the wavelet flux was in excellent agreement with eddy–



covariance flux, but for periods where the stationarity requirement was violated the wavelet flux was found to be more reliable and provided a better estimate. Since their work, wavelet analysis applied to analyze turbulent fluxes has become more common (von der Heyden et al., 2018; Göckede et al., 2019; Conte et al., 2021).

   The present work investigates an instrumented mobile car setup (shown in Fig. 1) by comparing car–based measurements with measurements made by a small roadside tripod. Our setup differs from Belušic et al. (2014) in two main

ways which are necessary to make the vehicle safe for on–road driving with other vehicles: (1) our sonic anemometer is held closer to the vehicle and situated over the vehicle's front end and (2) the sonic anemometer is held closer to the ground at a height of 1.7 m, which is near the height of the vehicle's top. We selected this design to investigate whether the sonic anemometer can be held closer to the vehicle and still provide measurements that are representative of the mean flow and turbulence near the surface, allowing road–safe vehicle operation without compromising the measured data. While farmland

is common in our measurement domain, the car also travelled past many large trees, houses, and experienced the occasional passing vehicle traveling in the opposite direction. Therefore, we investigate if the mobile car measurements are still representative of the turbulence statistics near the surface in a less idealized case, where the upwind surface and terrain is not homogenous, and the measured flow is affected by many surface obstacles including other traffic. Thus, this work aims to help design a low–cost experiment to measure and analyze on–road velocity variances and covariances using an instrumented car,

in the presence of sporadic passing traffic and upwind surface inhomogeneities. This study investigates how these inhomogeneities affects the calculated statistics. Wavelet analysis is considered as an alternative technique to eddy–covariance for the estimation of velocity variances and covariances measured on the car and is applied to quantify and remove the effects of sporadic passing traffic. The potential sources of measurement uncertainty on the car are quantified and discussed.

## 2 Methodology

### 2.1 Instrumented car

A sport utility vehicle (SUV) was outfitted with instrumentation fastened to the vehicle using a roof rack as shown in Fig. 1. A 40 Hz, three–dimensional sonic anemometer (Applied Technologies, Inc., model type "A" or "Vx") was installed on a support arm located at the front end of the vehicle, at a height of $z_m = 1.7$ m. Since the "A" type is rated for higher flow velocities, once it became available for use it was installed and the "Vx" type was removed. This change was done to test the

how the specific sonic anemometer model affects the measured velocities. The "A", "Vx", and "V" type sonic anemometers ("V" is used on the roadside tripod) have an accuracy of ± 0.1 m s$^{-1}$ within a measurement range of ± 60 m s$^{-1}$, ± 20 m s$^{-1}$, and ± 15 m s$^{-1}$ respectively. To limit the effect of vibrations on the measurements made by the sonic anemometer, the horizontal arm holding the anemometer was supported by two metal rods attached to the vehicle's front end. The forward scene was recorded by a Thinkware F750 dashcam (30 frames per second), which encodes 1 Hz measurements of latitude, longitude, and

vehicle speed ($s$) as metadata in each mp4 file.





The coordinate system of the sonic anemometer on the car is defined (assuming an observer is sitting inside of the vehicle facing toward the front hood) so that measured velocity parallel to vehicle motion ($u_m$) is positive toward the car, the measured lateral velocity ($v_m$) is positive toward the right, and the measured vertical velocity ($w_m$) is positive upward. Subscript $m$ denotes a raw measured value.

**2.2 Roadside tripod**

On 20 and 22 Aug 2019 a small tripod was assembled and placed at the roadside (i.e., in the highway shoulder) to compare with measurements made by the instrumented car as it travelled past the stationary tripod. The tripod was equipped with a three–dimensional sonic anemometer (Applied Technologies, Inc., model type "V") that recorded at either a frequency of 10 Hz (20 Aug) or 20 Hz (22 Aug). Each day the sonic was installed at a measurement height of $z_m = 1.4$ m. On 22 Aug, the tripod also had a Thinkware X700 dashcam (30 frames per second) installed to record passing traffic. To investigate the effect of tripod vibrations on the measurements, we tied down the system with string on the 22 Aug, but left it free to vibrate on the 20 Aug.

**2.3 Measurement site**

The measurement site was agricultural fields located on either side of a two–lane highway. The traffic on 22 Aug passing our measurement site was more significant than on 20 Aug; the traffic composition on 22 Aug included occasional large trucks and we did not observe any large trucks passing our measurement site on 20 Aug. Both days featured fair weather, with sky conditions ranging from mainly sunny on 20 Aug to partly cloudy on 22 Aug. The wind direction measured at nearby Egbert weather station (maintained by Environment and Climate Change Canada with measurements obtained at a height of 10 m) ranged between 160º and 200º on 20 Aug and 310º and 340º on 22 Aug. The mean wind ranged between 4.2 m s$^{-1}$ and 5.6 m s$^{-1}$ on 20 Aug and 3.8 m s$^{-1}$ and 5.0 m s$^{-1}$ on 22 Aug. The Egbert weather station is located about 16 km north of the measurement site.

The road is relatively flat near the tripod location, but in general, the terrain is not flat and homogenous in this area. The study area (which spans about 10 km) has several hills with slopes up to 10º. The elevation ranges between 200 and 300 m above mean sea level and there are areas with numerous trees and some structures located upwind of the highway. The tripod was located at an elevation of 277 m on 20 Aug, and at an elevation of 222 m on 22 Aug (estimated from Google Earth). For reference, the Egbert weather station is at an elevation of 251 m.

In this work a *measurement track* refers to the specific ground path driven by the vehicle, while a *measurement pass* refers to a specific set of measurements made on a particular track. Each measurement pass can be further divided into "A" and "B", representing the specific direction driven by the vehicle on a particular track. On each day, two different 1000 m tracks (Track #1 and Track #2) are chosen to compare with measurements made on the tripod. Track #1 is centered on the location of the tripod and consists of an equal amount of highway on either side of the tripod (i.e., 500 m before the tripod and 500 m after the tripod). Track #2 however begins 120 m away from the tripod and continues for 1000 m, thus it does not



include the highway directly in front of the tripod. Track #1 and Track #2 (for each day) are displayed in Fig. 2 as yellow and blue lines, respectively. The location of the tripod in Fig. 2 is displayed as a marker with a star enclosed. Track #1 and Track

#2 are chosen to examine how the choice of measurement track impacts the comparison of turbulence statistics between the car and tripod.

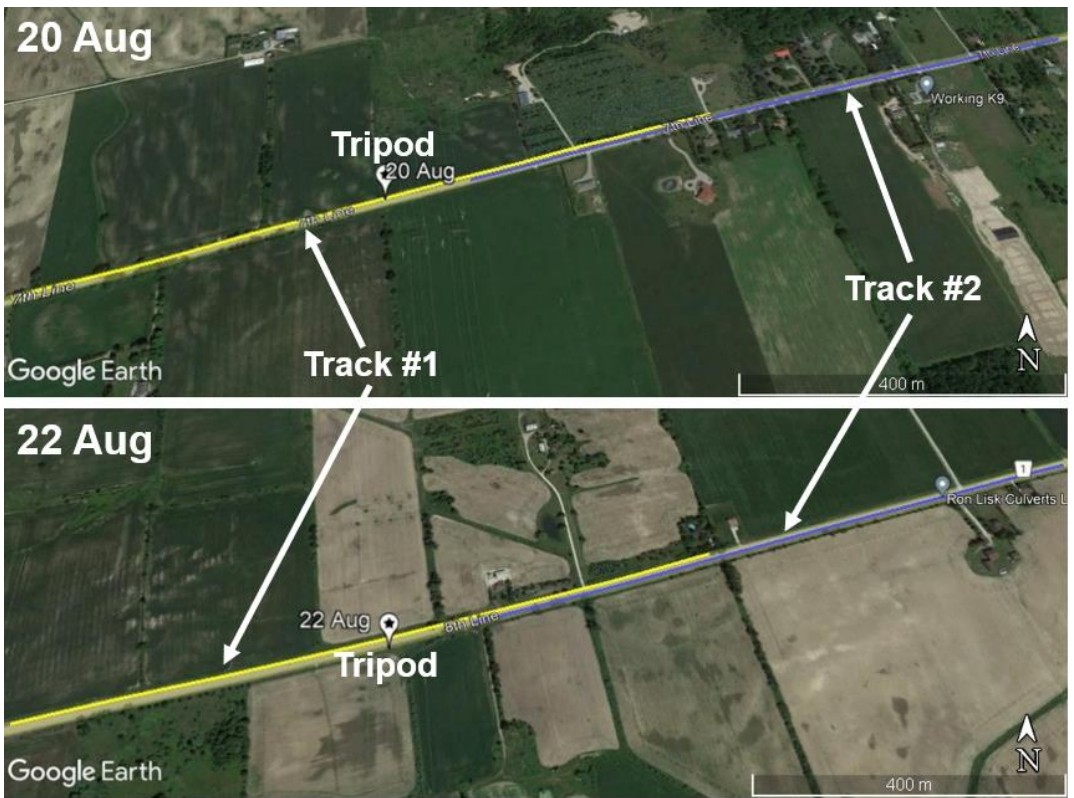

**Figure 2: The measurement site on 20 Aug (top) and 22 Aug (bottom) with the 1000 m tracks driven by the car superimposed. Track**
**#1 is shown as a yellow line and Track #2 is shown as a blue line. Track #1 is centered on the location of the tripod and therefore 500 m of highway is included in Track #1 on either side of the tripod location. Track #2 begins 120 m away from the tripod and therefore it does not include any measurements made on the highway directly in front of the tripod. © Google Earth Images.**

**2.4 Flow distortion and sensor corrections**

Measurements made on an instrumented car may be significantly impacted by flow distortion. Flow distortion originates from
vehicle movement (speed $s$) and from the ambient horizontal wind ($u_H$) that is present even when the vehicle is stationary; $u_H$ may be at an angle to the vehicle, potentially leading to flow distortion in both components of the measured horizontal velocity (i.e., $u_m, v_m$). Further impacts on the measurements can occur from sensor misalignment and sensor limitations that occur while measuring in high flow velocities. Flow distortion at the location of the sonic anemometer is investigated by analyzing measurement passes that are separated into part A and B. A and B are each driven on the same length of highway, but in
opposite directions (following Belusic et al. 2014). Before investigating flow distortion, the sonic anemometer data are filtered





for spikes. Here a spike is defined as an unrealistic sequence of 2 or less data points and is identified by applying a non–linear median filter according to Starkenburg et al. (2016). For the measurements considered in this paper, the effect of this spike removal on the calculated statistics is minimal (i.e., in any measurement pass there are 2 or less flagged values). Measurements flagged as spikes are removed and replaced with linearly interpolated values. If it is assumed that the mean ambient vertical

velocity $\bar{w} \approx 0$ m s$^{-1}$ and that the flow is in steady state during A and B with measurements made at a constant vehicle speed $s$ then following Belusic et al. (2014) and Miller et al. (2019), we can assume three relationships (here an uppercase variable ($U$, $V$, $W$, $S$) represents an averaged or binned value, while a lowercase variable represents an individual measurement).

I.    Without flow distortion, the average measured vertical velocity ($W$) at any measured longitudinal velocity ($U$) is
expected to be equal to 0, over a sufficiently long record. That is, $W$ is not expected to have any dependence on $U$. However, in the presence of flow distortion on the mobile car, $W$ becomes a function of $U$.

II.    The average velocity recorded over both travel directions ($U_{AB}$ as a function of $S$) is expected to follow the relationship $U_{AB}(S) = 0.5(U_A(S) + U_B(S)) = S$, since any wind component parallel to the direction of vehicle motion is cancelled out by travelling the same distance in both directions.

III.    The lateral velocity $V$ measured over all of A and all of B is expected to follow the relationship $V_{AB} = 0.5[V_A + V_B] = = 0$, since the coordinate system rotates 180° when the vehicle changes direction.

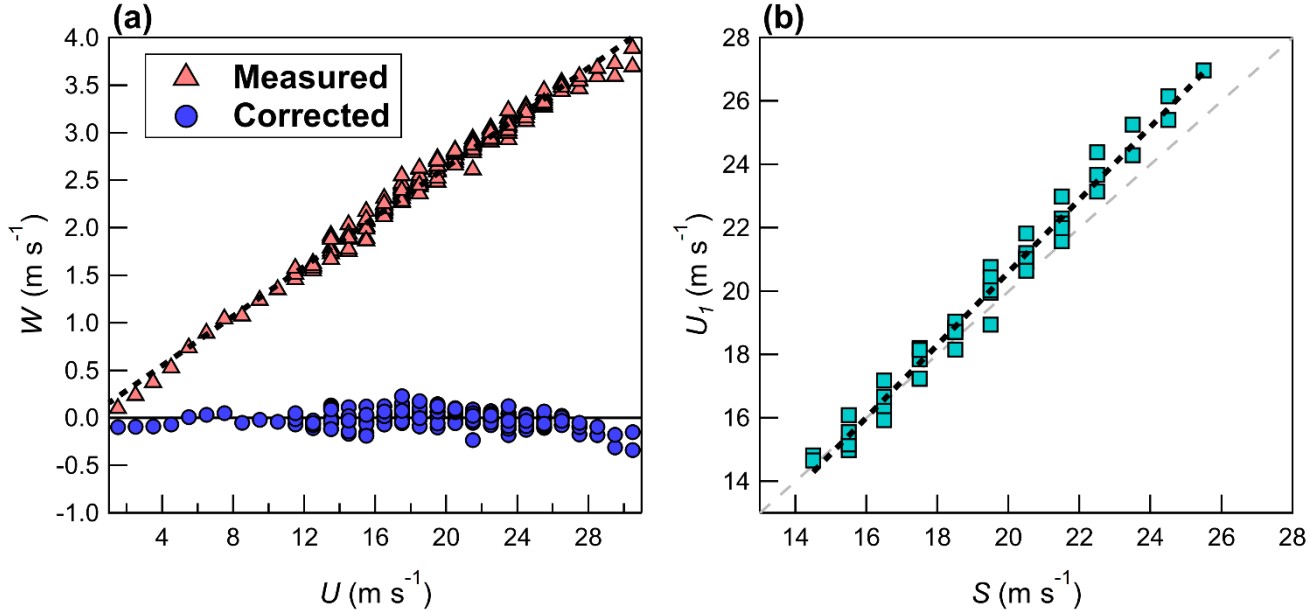

**Figure 3: (a) The measured vertical velocity $W$ (red) plotted as a function of the measured longitudinal velocity $U$ and the corrected**
**vertical velocity $W_c$ (blue) after application of Eq. (1); (b) the measured $U_1$ as a function of vehicle speed $S$ (after application of Eq. (1)). Measurements are binned using a bin size of 1 m s$^{-1}$. Data shown are for both 20 and 22 Aug. Black dashed lines give a least square fit: (a) $W = 0.03 + 0.13U$ ($R^2 = 0.99$) and (b) $U_1 = -2.34 + 1.147S$ ($R^2 = 0.98$).**



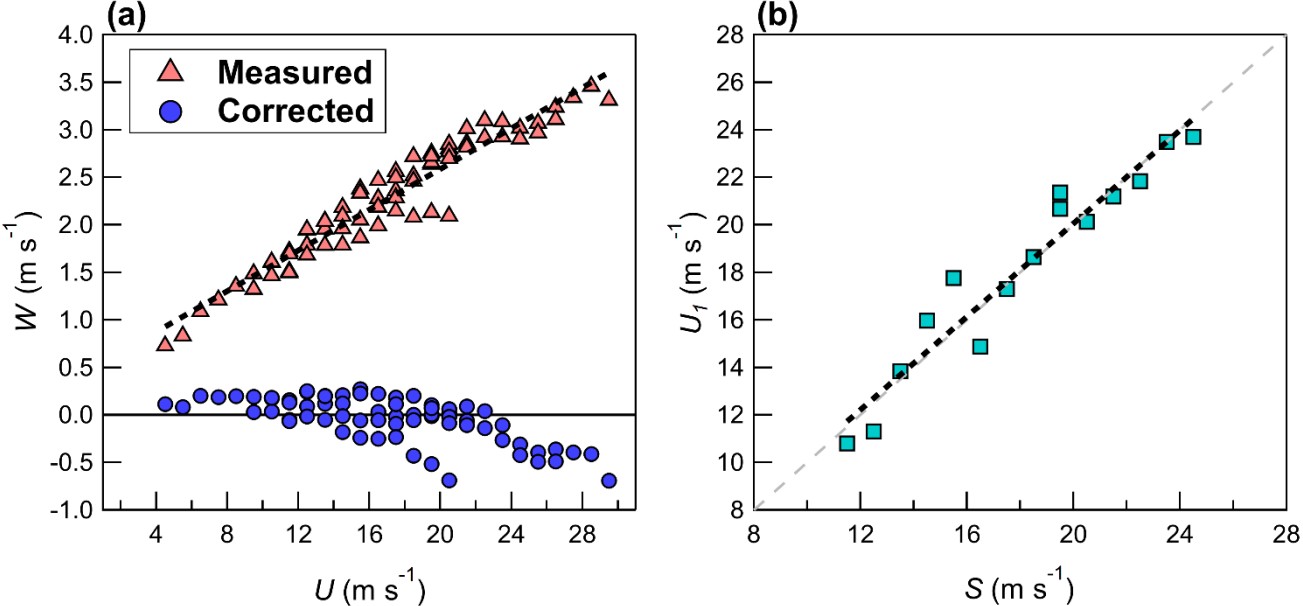

**Figure 4: Corrections as shown in Fig. 3, except for 30 Aug. Black dashed lines give a least square fit: (a) $W = 0.45 + 0.11U$ ($R^2 =$ 0.96) and (b) $U_1 = 0.44 + 0.98S$ ($R^2 = 0.92$).**

Figure 3(a) shows $W$ binned according to $U$, with binning completed using a bin size of 1 m s$^{-1}$. The data shown in Fig. 3 includes all back–and–forth passes completed on 20 and 22 Aug. The anemometer was not removed from the vehicle between 20 and 22 Aug, therefore the results should be consistent across both days. Figure 3(a) demonstrates that flow distortion at the measurement location is significant in this study, and $W$ increases linearly with increasing $U$ (coefficient of determination, $R^2$ = 0.99). The measured velocity field is corrected by applying a coordinate rotation to give a zero mean vertical velocity (assuming there is no flow distortion effect in $v_m$) as

$$u_1 = u_m \cos\theta + w_m \sin\theta, \tag{1 a}$$

$$w_c = -u_m \sin\theta + w_m \cos\theta. \tag{1 b}$$

Here $\theta$ is set to the median of $\theta_b$, where $\theta_b = \mathrm{atan}(W_b/U_b)$ and subscript $b$ represents individual binned values of 1 m s$^{-1}$ size (i.e., from Fig. 3(a)). $\theta_b$ does not show any dependence on $U$ for the vehicle speeds investigated in this study (i.e., for $S > 15$ m s$^{-1}$; see Fig. S1 in the supplementary material). For the data shown in Fig. 3(a), $\theta = 7.54°$ (interquartile range of 0.32°).

Figure 3(b) shows $U_1$ binned according to $S$. In Fig. 3(b) at $S > 17$ m s$^{-1}$ the results suggest that $U_1$ is overestimated. The same analysis performed on 30 Aug did not show this overestimation in $U_1$ for higher $S$ (Fig. 4b), however the setup on 30 Aug used a sonic anemometer that is rated for higher flow velocities, up to 60 m s$^{-1}$ (Applied Technologies, model "A"). This suggests that the overestimation in $U_1$ on 20 and 22 Aug for $S > 17$ m s$^{-1}$ is likely an instrument–related limitation rather than a direct effect of flow distortion. Taking the difference between the least square fit and the expected relationship (i.e., $U_1 = S$) the overestimation in $u_1$ (i.e., after applying Eq. (1)) is



$$u_{excess}(s) = \max(0, -2.34 + 0.147s). \tag{2}$$

The overestimation, $u_{excess}(s)$ is then removed from $u_1$ to give $u_c$ as

$$u_c(s) = u_1(s) - u_{excess}(s). \tag{3}$$

No corrections are applied to $v_m$ since there is no clear relationship with any measured variable (i.e., $U, S$; see Fig. S2). The corrections outlined in Eq. (1) through Eq. (3) are applied to all vehicle measurements from 20 and 22 Aug, for which $s > 0$ m s$^{-1}$. After correction for flow distortion the 1 Hz vehicle speed is linearly interpolated to 40 Hz and then removed from $u_c$ to give the meteorological wind speed component parallel to the direction of motion as (Belusic et al., 2014)

$$u = u_c - s. \tag{4}$$

**2.5 Wavelet analysis and the quantification of sporadic passing traffic**

The continuous wavelet transform of a discrete time series $x$ containing $N$ data points, measured at a time step $\Delta t$ is calculated as (Torrence and Compo, 1998)

$$G_n^x(a) = \sum_{n'=0}^{N-1} x_{n'} \psi_0^* \left( \frac{(n'-n)\Delta t}{a} \right). \tag{5}$$

The wavelet coefficients are calculated as the convolution of $x$ with a dilated ($a$) and translated ($n$) wavelet function $\psi_0$, where $a$ is referred to as the wavelet scale and $n$ is a localized time (position) index. If $\psi$ is complex, then the complex conjugate ($*$) is used to calculate $G_n^x(a)$. Following Torrence and Compo, (1998), the analyzing wavelet is normalized to have unit energy, so that

$$\psi_0 = \sqrt{\frac{\Delta t}{a}} \psi. \tag{6}$$

Where $\psi$ in this work is the complex Morlet wavelet,

$$\psi(\eta) = \pi^{-0.25} e^{6i\eta} e^{-\eta^2/2}. \tag{7}$$

The Morlet wavelet is chosen since it has been shown to be well suited for the analysis of atmospheric turbulence (Strunin and Hiyama, 2004; Salmond 2005; Schaller et al., 2017). The total energy (or wavelet variance) of the entire time series is preserved in the wavelet transform and can be recovered by summing the scaled–averaged wavelet power over all scales ($j$) and times ($n$),

$$\sigma_x^2 = \frac{\Delta j \Delta t}{C_\delta N} \sum_{n=0}^{N-1} \sum_{j=0}^{J} \frac{1}{a_j} \left| G_n^x(a_j) \right|^2, \tag{8}$$

where $\Delta j = 0.25$ determines the spacing between discrete scales $a_j = a_0 2^{j\Delta j}$ ($a_0 = 2\Delta t$) and $C_\delta = 0.776$ is a wavelet specific reconstruction factor for the Morlet wavelet. The Morlet wavelet scale can be converted to an equivalent Fourier scale (i.e.,



period) as $\tilde{a}_j = 1.03 a_j$. Like the wavelet variance, given time series $x_n$ and $y_n$, the wavelet covariance (or turbulent flux) can

be calculated as

$$\overline{x'y'} = \frac{\Delta j \Delta t}{C_\delta N} \sum_{n=0}^{N-1} \sum_{j=0}^{J} \frac{1}{a_j} \, \Re[G_n^x(a_j) G_n^{y*}(a_j)], \tag{9}$$

where the real part ($\Re$) of the wavelet cross–spectrum defines the wavelet co–spectrum, and the imaginary part gives the wavelet quadrature spectrum (Strunin and Hiyama, 2004; Paterna et al., 2016). For a 1000 m track consisting of $N_T = T/\Delta t$ ($T \in \mathbb{Z}$ is the integer second length of the track) measurements the wavelet variance including time scales up to index $a^*$ can

be calculated as

$$\sigma^2_{x \, 1 \, \text{km}} = \frac{\Delta j \Delta t}{C_\delta N_T} \sum_{n=0}^{N_T-1} \sum_{j=0}^{a^*} \frac{1}{a_j} \, \left| G_n^x(a_j) \right|^2. \tag{10}$$

In Eq. (10) index value $a^*$ represents the maximum wavelet scale, which is set to match $T$ as closely as possible. $G_n^x(a_j)$ is calculated from a time series with a temporal length 11 times that of $\sigma^2_{x \, 1 \, \text{km}}$, and the times corresponding to $\sigma^2_{x \, 1 \, \text{km}}$ are at the center of the period. This approach is applied to ensure that the wavelet transform coefficients used to calculate the wavelet

variances are not impacted by edge effects for scales up to $a^*$ (i.e., they do not lie outside of the cone of influence), while still retaining good computational efficiency (Torrence and Compo, 1988; Schaller et al., 2017). $\sigma^2_{x \, 1 \, \text{km}}$ can be decomposed to give the wavelet variance for each second of the track (likewise with scales up to index $a^*$), as

$$\sigma^2_{x_{i \, 1 \, s}} = \frac{\Delta j \Delta t}{C_\delta N} \sum_{n=i/\Delta t}^{(i+1)/\Delta t - 1} \sum_{j=0}^{a^*} \frac{1}{a_j} \left| G_n^x(a_j) \right|^2, \tag{11}$$

where $i = 0, 1, \ldots, T-1$, $N = 1/\Delta t$ and

$$\sigma^2_{x \, 1 \, \text{km}} = \frac{1}{T} \sum_{i=0}^{T-1} \sigma^2_{x_{i \, 1 \, s}}. \tag{12}$$

The wavelet variance calculated for each second allows the effects of sporadic passing traffic to be removed by excluding times when traffic is likely affecting the measurements made on the car (as determined by manual inspection of the video recordings), calculated as

$$\sigma^2_{x \, F \, 1 \, \text{km}} = \frac{1}{T_f} \sum_{i=0}^{T-1} \delta_i \sigma^2_{x_{i \, 1 \, s}}, \tag{13}$$

where $\delta_i = \begin{cases} 0, & \text{if traffic} \\ 1, & \text{otherwise} \end{cases}$ and $T_f = \sum_{i=0}^{T-1} \delta_i$.

Using the real part of the wavelet coefficients, the original time series $x$ can be reconstructed at each $n$. By limiting the scales (for example selecting scales $j = J_{\min}$ to $j = J_{\max}$) a wavelet filtered time series can be constructed at each $n$ as

$$x_n^f = \frac{\Delta j \sqrt{\Delta t}}{C_\delta \psi_0(0)} \sum_{j=J_{\min}}^{J_{\max}} \frac{1}{\sqrt{a_j}} \Re[G_n^x(a_j)], \tag{14}$$



where $\psi_0(0) = \pi^{-0.25}$ for the Morlet wavelet. Calculation of the wavelet transform is computationally intensive when Eq. (5)

is used. By applying the convolution theorem, the wavelet transform can be completed much faster in Fourier space and this

approach is used here; the software developed to perform the continuous wavelet transform has been converted to IGOR Pro

from Matlab code available online by Torrence and Compo (1998).

**2.6 Coordinate rotation**

To compare the measurements made on the tripod to those made on the car, the coordinate systems must be consistent. The

initial step is to rotate the individual measurements made on the vehicle into a meteorological coordinate system (i.e., $u_{met}$

positive toward the east and $v_{met}$ positive toward the north) using the vehicle's heading. This rotation is necessary since the

vehicle's heading may change along the measurement path, leading to a varying sonic anemometer coordinate system along a

driven path. For driven paths with large curvature, not performing the transformation to meteorological coordinates gives

incorrect mean values (and variances) that are used to determine the rotation angles needed for transformation into a streamwise

coordinate system. For the highways investigated in this study, the vehicle heading remains rather consistent over their length;

hence our analysis only applies to straight vehicle motion, and we do not determine uncertainties due to measurements through

road curvature.

After rotation into meteorological coordinates, each track (on the car and tripod) is then rotated into a mean

streamwise coordinate system following Wilczak et al. (2001), where $\bar{u}$ is the mean wind and $\bar{v} = \bar{w} = 0$. The wavelet

variances and covariances are likewise rotated into mean streamwise coordinates (unless otherwise indicated) using the same

rotation angles applied to rotate the eddy–covariance results.

**2.7 Sampling errors**

**2.7.1 Random measurement uncertainty**

For the calculation of turbulence statistics, the use of a finite record length gives rise to a random measurement uncertainty

since the record will not contain enough independent samples to accurately represent the ensemble mean (Lenschow et al.,

1994). Further random measurement uncertainty can be introduced by non–stationarity in the record and white noise in the

measured signal (Rannik et al., 2016). In this work, the magnitude of the random measurement uncertainty is estimated using

two methodologies. All uncertainty estimations are after correction for flow distortion and rotation into a streamwise

coordinate system. The first method developed by Mann and Lenschow (1994) can be defined as

$$\delta_{ML} = |\overline{w'q'}| \left(\frac{2I_{wq}}{T}\right)^{\frac{1}{2}} \left(\frac{1+r_{wq}^2}{r_{wq}^2}\right)^{\frac{1}{2}} (1 - az^*),$$                                           (15)

with the integral time scale ($I_{ws}$) calculated as

$$I_{wq} = \int_0^\infty R_{wq}(\tau)\, d\tau.$$                                                                                                                                     (16)




$I_{wq}$ is estimated by numerically integrating the autocorrelation function to the first zero crossing. In Eq. (15) $z^* \cong 0$ near the surface, $r_{wq} = \frac{\overline{w'q'}}{\sigma_w \sigma_q}$ is the correlation coefficient between $w$ and $q$, and $T$ is the averaging period (in seconds) over which the

covariance is calculated. For neutral stability, $I_{wq}$ can be approximated as $z/s$ (Finkelstein and Sims, 2001). For a vehicle with a measurement height of $z_m = 1.7$ m, a mean wind speed of $\bar{u} \approx 2.5$ m s$^{-1}$, and a constant vehicle speed of $s = 25$ m s$^{-1}$, $I_{wq} \approx$ 0.07 s. For the stationary tripod ($s = 0$) at a slightly lower height of $z_m = 1.4$ m, $I_{wq} = 0.56$ s for the same wind speed.  For a covariance of scalar $q$ with the vertical velocity $w$, the instantaneous flux is calculated as $\varphi' = w'q' = (w - \bar{w})(q - \bar{q})$ and $\varphi'$ is used to estimate the autocorrelation function needed for calculation of the integral time scale ($I_{wq}$) from Eq. (16) (Rannik

et al., 2016). The instantaneous flux is introduced since the cross–correlation is an asymmetric function making it unsuitable for estimation of the $I_{wq}$.

The second methodology outlined in Finkelstein and Sims (2001) gives an estimation of the variance of a covariance ($\delta_{FS}$),

$$\delta_{FS} = \sqrt{var(\overline{w'q'})} = \left[\frac{1}{N}\left(\sum_{p=-m}^{m} \hat{\gamma}_{q,q}(p)\hat{\gamma}_{w,w}(p) + \sum_{p=-m}^{m} \hat{\gamma}_{q,w}(p)\hat{\gamma}_{w,q}(p)\right)\right]^{1/2}, \qquad (17)$$

where $m$ is the number of samples required to ensure the integral time scale (ITS) is sufficiently captured. $\hat{\gamma}_{w,w}(p)$ and $\hat{\gamma}_{w,q}(p)$ are the unbiased autocovariance and cross–covariance respectively, expressed as

$$\hat{\gamma}_{w,w}(p) = \frac{1}{N-p}\sum_{i=1}^{N-p}(w_i - \bar{w})(w_{i+h} - \bar{w}), \qquad (18)$$

and

$$\hat{\gamma}_{q,w}(p) = \frac{1}{N-p}\sum_{i=1}^{N-p}(q_i - \bar{q})(w_{i+h} - \bar{w}). \qquad (19)$$

The value of $m$ is determined by calculating $\delta_{FS}$ as a function of $m$ and choosing the value at which $\delta_{FS}$ reaches a constant or asymptotic value as $m$ is further increased. For the roadside tripod a value of $m = 300$ s is determined, while for the vehicle measurements $m = 30$ s (see Fig. S4 and Fig. S5)

For wavelet analysis, Eq. (14) is applied to generate a wavelet reconstructed time series ($q_f$ and $w_f$) for scales up to $a^*$. Thus, the reconstructed time series will exclude low frequency contributions attributed to wavelengths $\lambda > 1000$ m. The

reconstructed time series are then rotated into mean streamwise coordinates and subsequently used in Eq. (18) and Eq. (19) to estimate $\delta_{FS}$ for the wavelet covariance (and likewise for wavelet variances).

**2.7.2 Random measurement uncertainty due to instrument noise only**

The sonic anemometer's signal may be impacted by white noise, a form of random measurement uncertainty. Lenschow et al. (2000) consider a stationary time series with its mean removed (i.e., $w'(t)$) that is impacted by (uncorrelated)

white noise, $\epsilon(t)$, where the autocovariance function is





$$\gamma_{w,w}(\tau) = \overline{(w' + \epsilon')(w'_{t+\tau} + \epsilon'_{t+\tau})} = \overline{w'w'_{t+\tau}} + \overline{w'\epsilon'_{t+\tau}} + \overline{w'_{t+\tau}\epsilon} + \overline{\epsilon'\epsilon'_{t+\tau}}. \tag{20}$$

Since $w(t)$ and $\epsilon(t)$ are uncorrelated, $\epsilon(t)$ is present only at zero lag and so $\overline{w\epsilon} = 0$. Equation (20) then reduces to $\gamma_{w,w}(\tau) = \overline{w'w'_{t+\tau}}$, with $\gamma_{w,w}(0) = \overline{w'^2} + \overline{\epsilon'^2}$. Based on the inertial subrange theory by Kolmogorov, the autocovariance function expected to follow (Lenschow et al., 2000; Wulfmeyer et al., 2010; Bonin et al., 2016),

$$\gamma_{w,w}(\tau) = \overline{w'^2} - C\tau^{\frac{2}{3}}, \tag{21}$$

where constant $C$ is associated with turbulent eddy dissipation. To estimate $\overline{\epsilon'^2}$, Eq. (21) is typically fit to the first 5 lags of the autocovariance function, corresponding to time lags of 0.1 to 0.5 s for a 10 Hz signal of a sonic anemometer (Rannik et al., 2016). For Doppler lidar measurements of the vertical velocity in convective conditions, Bonin et al. (2016) fit Eq. (21) to the autocovariance function for time lags up half the integral time scale (i.e., $\tau = 0.5I_{ww}$). The fit is then extrapolated back to zero

lag to give $\gamma_{w,w}(\to 0)$, and the variance attributed to white noise in the measured signal is then estimated as (Lenschow et al. 2000; Mauder et al., 2013)

$$\overline{\epsilon'^2} = \Delta\gamma_{w,w} = \gamma_{w,w}(0) - \gamma_{w,w}(\to 0). \tag{22}$$

Some authors report a poor fit to Eq. (21) and instead apply a linear fit extrapolation back to zero lag to determine Eq. (22) (Lenschow et al., 2000; Mauder et al., 2013; Langford et al., 2015). For measurements obtained on the tripod and

instrumented car, a linear fit extrapolation in addition to Eq. (21) are used to estimate $\overline{\epsilon'^2}$. For tripod measurements time lags up to 0.5 s are used to determine the fit, but for the car travelling at vehicle speeds near 20 m s$^{-1}$, only the first 3 points (up to 0.075 s) of the autocovariance function are used. Eq. (21) may lead to an extrapolated value at zero lag larger than $\gamma_{w,w}(0)$, which gives 'negative' and thus undefined $\overline{\epsilon'^2}$. Bonin et al. (2016) noted a similar finding in their investigation when fitting the autocovariance function to Eq. (21) for the vertical velocity measured from Doppler lidar. They hypothesize that the

undefined $\overline{\epsilon'^2}$ occurs when the genuine white noise in the signal is minimal and the smallest scales of turbulence remain unresolved. Therefore, when $\overline{\epsilon'^2}$ is negative and undefined we assume that the true white noise is minimal and that $\overline{\epsilon'^2} \approx 0$. Thus, for the analysis herein $\overline{\epsilon'^2} = \max(\overline{\epsilon'^2}, 0)$.

**2.8 Comparison of mobile car measurements to tripod measurements**

In this work, we follow the approach of Belusic et al. (2014) and select a fixed ground path to investigate means, variances

and covariances on the car. Two different fixed 1000 m ground paths ($L$) are considered, referred to as Track #1 and Track #2, and these tracks are compared to measurements made by the tripod (see Sect. 2.3).

The averaging period ($T$) on the tripod is set to 5 min for atmospheric means, but for atmospheric variances and covariances $T$ varies depending on the mean wind speed measured by the tripod ($\bar{u}$) according to Taylor's frozen hypothesis as $T = L/\bar{u}$, where $L = 1000$ m. For the two measurement days investigated here, $T$ on the tripod ranges between 6 and 8 min.

For consistency, the averaging period used for calculation of the tripod means, variances and covariances is centered on the



time that the instrumented car passes the tripod (for both Track #1 and Track #2). For the car, any measurement pass that follows closely behind a vehicle is excluded from the results.

## 3.0 Results and discussion

### 3.1 Mean wind speed and mean wind direction

Figure 5 shows a scatter plot of (a) the 5 min mean wind direction on the tripod compared to the mean wind direction measured on the mobile car and (b) the 5 min mean wind speed measured on the tripod compared to the mean wind speed measured on the mobile car. The mean wind speed shown is after rotation into streamwise coordinates. The grey lines in Fig. 5 denote a specific percentage of the tripod measured value (i.e., 100% gives a one–to–one relationship) and this convention is used in the figures that follow. The mean bias error, MBE $= (1/N) \sum_{i=1}^{N} (M_c - M_t)$, and the root mean squared error, RMSE $=$

$(1/N) \sum_{i=1}^{N} \sqrt{(M_c - M_t)^2}$ are given in Table 1. Here, the subscripts $c$ and $t$ refer to the car and the tripod. The tripod is therefore used as a 'ground truth' for the car measurements.

**Table 1: Statistics calculated over all measurement passes (i.e., on both tracks on 20 and 22 Aug). Subscript $EC$ denotes a statistical variance or a covariance calculated using eddy–covariance. A subscript $W$ denotes a variance or covariance calculated using wavelet**
**analysis.**

| | $MBE_{EC}$ | $MBE_W$ | $RMSE_{EC}$ | $RMSE_W$ | $Mean_{EC}$ Car | $Mean_W$ Car | $Mean_{EC}$ Tripod |
|---|---|---|---|---|---|---|---|
| $\overline{u'^2}$ (m$^2$ s$^{-2}$) | 0.90 | 0.44 | 1.44 | 0.75 | 2.15 | 1.69 | 1.26 |
| $\overline{v'^2}$ (m$^2$ s$^{-2}$) | 0.20 | 0.04 | 0.61 | 0.44 | 1.38 | 1.21 | 1.19 |
| $\overline{w'^2}$ (m$^2$ s$^{-2}$) | −0.11 | −0.12 | 0.12 | 0.13 | 0.17 | 0.16 | 0.29 |
| $\overline{u'w'}$ (m$^2$ s$^{-2}$) | 0.005 | 0.02 | 0.08 | 0.08 | −0.13 | −0.11 | −0.14 |
| $\overline{w'T'}$ (K m s$^{-1}$) | −0.05 | −0.04 | 0.06 | 0.06 | 0.08 | 0.08 | 0.13 |
| $\bar{u}$ (m s$^{-1}$) | 0.04 | | 0.53 | | 2.45 | | 2.42 |


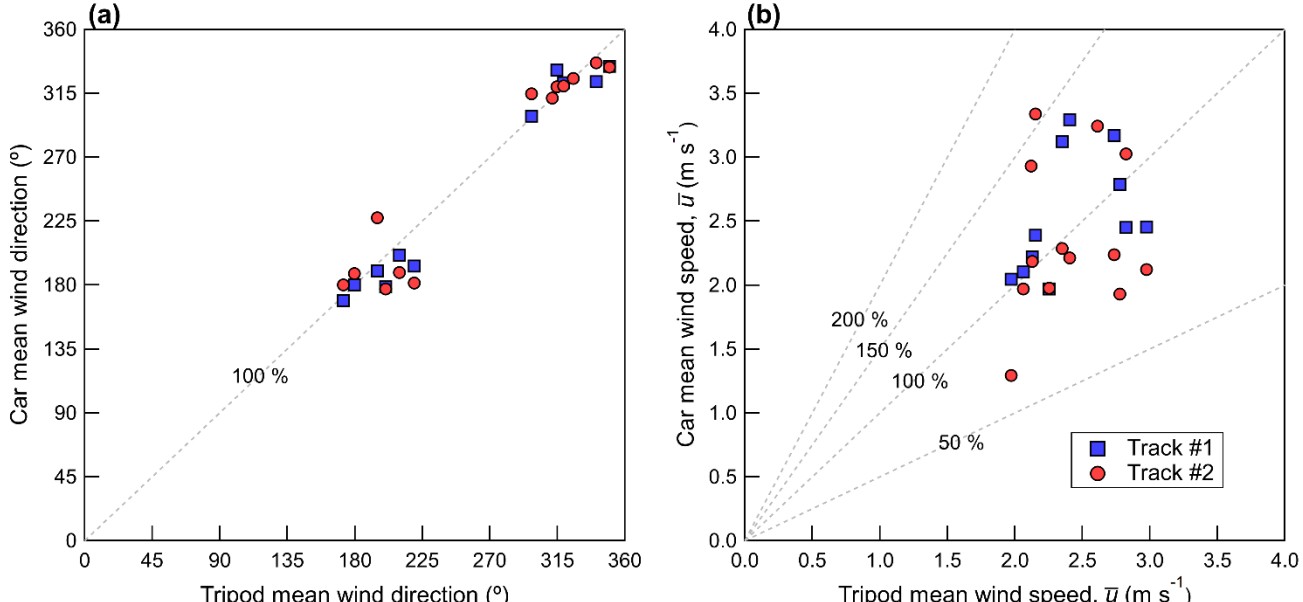

**Figure 5: A scatter plot showing the mean wind direction (a) and mean wind speed (b) measured by the tripod and compared to the mobile car. Dashed grey lines denote constant percentages of the independent variable.**

The mean wind speed shown in Fig. 5(b) shows relatively good agreement between the car and tripod for (car $\bar{u}$ = 2.45 m s$^{-1}$ compared to tripod $\bar{u}$ = 2.42 m s$^{-1}$ and car RMSE = 0.53 m s$^{-1}$). When the analysis is separated by tracks, the agreement is best for Track #1; RMSE = 0.43 m s$^{-1}$ and 0.71 m s$^{-1}$ for Track #1 and Track #2 respectively (see Table S1 and Table S2). If $\bar{u}$ measured on the tripod is used as a normalizing factor, the normalized root–mean squared error of $\bar{u}$ (NRMSE) is 18% and 30% for Track #1 and Track #2 respectively. The mean wind direction on the car agrees well with the tripod on both Track #1 and Track #2 as shown in Fig. 5(a), where most points fall within 20° of the one–to–one line.

To investigate how the car performs for shorter averaging periods, non–overlapping intervals of 10 s duration are examined on 20 and 22 Aug. There are 263 and 250 such intervals on 20 and 22 Aug respectively and these represent times that the vehicle is driving in the vicinity of the tripod (i.e., within about 10 km) and not necessarily on a 1000 m track. The results are shown in Table 2, which displays the average meteorological wind components ($u_{met}$ and $v_{met}$), the mean wind direction, and the mean wind speed (after rotation into streamwise coordinates). Statistics are also shown in Table 2, including the median, maximum and minimum values in each set and the interquartile range (IQR). The standard deviation of the wind direction is calculated using the Yamartino algorithm (Turner, 1986). The results show that the wind direction is rather consistent on both days for a shorter averaging period of 10 s, where the wind direction standard deviation is 38° on 20 Aug and 31° on 22 Aug. While the average of all 10 s mean wind speeds on 20 and 22 Aug is consistent with the measurement passes shown in Fig. 5(b), there can be significant variation in each individual interval as demonstrated by the large IQR and maximum/minimum values (IQR = 1.30 m s$^{-1}$ and 1.86 m s$^{-1}$ on 20 and 22 Aug respectively). This demonstrates that using



short averaging periods on the mobile car allows measurement of localized flow variations, where the magnitude of the flow may vary significantly but the direction remains relatively constant in comparison.

**Table 2: Statistics of the mean flow measured by the car on 20 and 22 Aug. The averaging period is 10 s; therefore, the statistics are calculated from a set of $n$ non–overlapping intervals. Shown are the wind components in a meteorological coordinate system ($u_{met}$, $v_{met}$), the mean wind direction calculated from $u_{met}$ and $v_{met}$, as well as the mean wind speed after rotation into a streamwise coordinate system. Note that $\overline{u}$ includes a component due to the vertical velocity, and hence it may exceed the horizontal wind speed calculated as $u_h = \sqrt{u_{met}^2 + v_{met}^2}$. The standard deviation of the wind direction is calculated using the Yamartino algorithm (Turner,** 
**1986).**

| | 20 Aug ($n = 263$) | | | | | | 22 Aug ($n = 250$) | | | | | |
|---|---|---|---|---|---|---|---|---|---|---|---|---|
| | Mean | Std Dev | Max | Min | Median | IQR | Mean | Std Dev | Max | Min | Median | IQR |
| $u_{met}$ (m s⁻¹) | 0.30 | 0.82 | -- | -- | 0.26 | 1.07 | 1.37 | 0.96 | -- | -- | 1.29 | 1.35 |
| $v_{met}$ (m s⁻¹) | 1.63 | 1.03 | -- | -- | 1.56 | 1.41 | −1.73 | 1.26 | -- | -- | −1.64 | 1.75 |
| $\theta$ (°) | 190 | 38.2 | -- | -- | 191 | 41.9 | 322 | 31.0 | -- | -- | 318 | 34.0 |
| $\overline{u}$ (m s⁻¹) | 1.90 | 0.92 | 4.54 | 0.18 | 1.72 | 1.30 | 2.42 | 1.23 | 6.30 | 0.14 | 2.40 | 1.86 |

### 3.2 Velocity variances and covariances

Figure 6 shows the velocity variances measured on the instrumented car compared to the velocity variances measured on the tripod. Figure 6(a), (b) and (c) show $\overline{u'^2}$, $\overline{v'^2}$ and $\overline{w'^2}$ respectively. The velocity variances measured on the car are calculated

using the typical statistical approach, denoted as EC (i.e., for time series $x$ with $N$ points, $\sigma_x^2 = (1/N) \sum_{i=1}^{N} (x - \bar{x})^2$) or wavelet analysis (i.e., Eq. (10)). Only statistical velocity variances measured by the tripod (and covariances calculated using eddy–covariance) are presented herein. For measurements made on the tripod the effect of applying wavelet analysis to calculate variances and covariances is minimal compared to the instrumented car (see Fig. S3). Furthermore, for some measurement passes the Morlet wavelet applied to the tripod suffers from edge effects that cannot be avoided, since the tripod

recordings were abruptly ended at the end of each measurement day. For wavelet analysis, the maximum wavelet scale (index $a^*$) is chosen to correspond as closely as possible to the temporal length of the measurement track to ensure that both calculation methods retain the same spatial scales and are therefore comparable (see Sect 2.5). For the car measurement tracks investigated here, the temporal length ranges between 40 and 60 s, and all measurement tracks have a maximum spatial scale of approximately 1000 m.



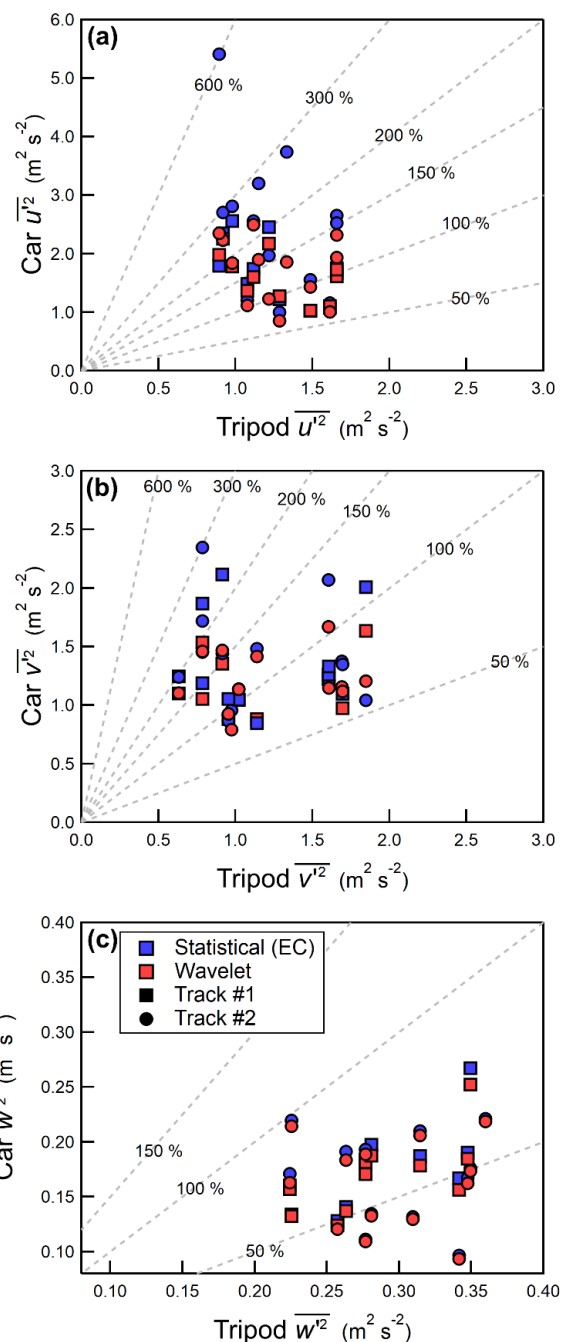

**Figure 6: The horizontal streamwise velocity variance, $\overline{u'^2}$ (a), the lateral velocity variance, $\overline{v'^2}$ (b) and the vertical velocity variance, $\overline{w'^2}$ (c) measured by the tripod (horizontal) and compared to the instrumented car (vertical). Variances calculated using either wavelet analysis or EC and are shown as red and blue markers respectively. Dashed grey lines denote constant percentages of the independent variable.**





Applying wavelet analysis to estimate the horizontal velocity variances leads to a significant reduction in the magnitude compared to EC for some passes, specifically for those passes reporting the largest horizontal velocity variances as shown in Fig. 6(a) and (b). This reduction results in an improved agreement between the two measurement systems; for $\overline{u'^2}$, wavelet analysis gives $\mathrm{RMSE_W} = 0.75$ m$^2$ s$^{-2}$ compared to $\mathrm{RMSE_{EC}} = 1.44$ m$^2$ s$^{-2}$ for EC. However, retaining larger scales in

the wavelet variance calculation (i.e., corresponding to spatial scales exceeding 1000 m) gives horizontal velocity variances that are larger and more consistent with EC. This suggests that wavelet analysis can better resolve low frequency variations occurring at spatial scales near and exceeding 1000 m, compared to EC. Low frequency contributions on the car may arise from variation in the flow that results only from a changing upwind environment, and therefore this effect would not be captured by a stationary monitoring station. As discussed in Sect 2.5, wavelet analysis is applied to a time series with a temporal

length 11 times longer than the time series used to calculate the EC variances, giving wavelet analysis superior low frequency resolution compared to eddy–covariance.

Despite the improved agreement when wavelet analysis is applied to estimate the horizontal velocity variances, there are still instances where $\overline{u'^2}$ and $\overline{v'^2}$ measured by the mobile car are larger than what is measured by the roadside tripod. Given the public highway where the study was conducted, some measurement passes inevitably have sporadic traffic that was

travelling in the opposite direction as the mobile car (as determined by visual inspection of the video). The passing traffic can significantly impact the velocity variances measured on the car due to vehicle–induced turbulence, especially in the case of passing heavy–duty trucks (Gordon et al., 2012; Miller et al., 2019). For the measurement passes shown in Fig. 6, there are two instances where a heavy–duty truck travelled in the lane opposite to the instrumented car, as well as a few occasions where passenger vehicles (i.e., SUV, cars) travelled past the car.

Figure 7 displays the 1 s wavelet variance calculated using Eq. (11) for three different measurement passes from Track #2 (on 22 Aug); Fig. 7(a) had 2 simultaneous passing sport utility vehicle (SUV), and Fig. 7(c) had a passing heavy–duty truck followed in quick succession by an SUV. Wavelet analysis is performed on the measured velocities in a meteorological coordinate system (i.e., $u_{met}, v_{met}$), with $a^*$ extending up the temporal length of the measurement pass (i.e., the same $a^*$ used for the wavelet variances presented in Fig. 6). Each measurement pass shown in Fig. 7 was performed in the

same direction and in the highway lane closest to the tripod (i.e., on the downwind side of the highway). Traffic is denoted by a circled area in the respective figure panel. With these instances of traffic included, the velocity variances are 1.68 m$^2$ s$^{-2}$, 1.38 m$^2$ s$^{-2}$ and 0.21 m$^2$ s$^{-2}$ for $\overline{u_{met}'^2}$, $\overline{v_{met}'^2}$ and $\overline{w_c'^2}$ respectively. Removing the 1 s wavelet variances corresponding temporally with these passing vehicles (9 seconds in total), gives a $\overline{u_{met}'^2}$, $\overline{v_{met}'^2}$ and $\overline{w_c'^2}$ of 1.47 m$^2$ s$^{-2}$, 1.29 m$^2$ s$^{-2}$ and 0.17 m$^2$ s$^{-2}$ respectively, representing about a 10% reduction in the turbulent kinetic energy during this measurement pass. This demonstrates that even

limited traffic travelling in the highway lane adjacent to the car (and in the opposite direction) can substantially increase the magnitude of the velocity variances measured by the car on a 1000 m track, especially heavy–duty trucks. In Fig. 7(a), two SUVs passed by the mobile car in quick succession, but the passage of these vehicles is not discernable as a localized increase





of the 1 s wavelet variances. This suggests that the vehicle wakes did not advect past the instrumented car during this measurement pass, and thus no removal is warranted.


**Figure 7: Car–measured velocity variances on 3 different 1000 m tracks calculated every second using wavelet analysis. The data shown are from 22 Aug. The black circled areas denote the passage traffic in the lane adjacent to the instrumented car (i.e., traveling in the opposite direction), as determined from visual inspection of the dashcam video. The text located to the right of the circle gives the traffic composition. The data shown are measurements from the lane closet to the tripod. The velocity variances shown are in a meteorological coordinate system.**







For the measurement pass shown in Fig. 7(b) there is a noticeable increase in the 1 s horizontal velocity variances about 450 m into the measurement track. A similar trend is also seen in Fig. 7(c). Before 450 m there are many large trees and houses upwind of the highway, but after 450 m the upwind environment becomes open farmland (i.e., limited obstructions to the mean flow). The presence of many trees and houses in close proximity acts as a windbreak, forcing the flow to accelerate

and rise over the surface obstructions. The flow is reduced downwind of the surface obstruction (Taylor and Salmon, 1993; Mochida et al., 2008), and close to the surface just after the obstruction (i.e., the near wake) is the "quiet zone", where the horizontal velocity variances are reduced in comparison with the undisturbed upwind flow (Lee and Lee, 2012; Lyu et al., 2020). Therefore, the reduced horizontal velocity variances for the first few hundred meters of the track may be related to the quiet zone generated by the many trees and houses upwind of the road. After about 450 m the upwind environment becomes

relatively open and the flow measured on the car increases, and this increase continues over the remainder of the track. The changing wind speed along the track introduces a trend in the horizontal velocity record measured on the car.

Figure 6(c) displays $\overline{w'^2}$ measured on the mobile car compared to $\overline{w'^2}$ measured on the tripod. The instrumented car $\overline{w'^2}$ is biased low by 30 to 50% ($\mathrm{MBE_{EC}} = -0.11 \, \mathrm{m^2 \, s^{-2}}$) and applying wavelet analysis to estimate $\overline{w'^2}$ does not improve the agreement between the two measurement systems. The removal of vehicle–induced turbulence from the car measurements

(and not the tripod) further decreases $\overline{w'^2}$, in turn increasing the bias between the car and tripod.

Figure 8(a) displays the vertical momentum flux $(\overline{u'w'})$ and Fig. 8(b) shows the sonic heat flux $(\overline{w'T'})$. Figure 8 follows the same conventions as Fig. 6. Like $\overline{w'^2}$, the sonic heat flux $(\overline{w'T'})$ measured by the mobile car in this study also has a low bias by 30 to 50% compared to the tripod ($\mathrm{MBE_{EC}} = -0.05 \, \mathrm{K \, m \, s^{-1}}$). There is no improvement in the statistical measures if wavelet analysis is used to estimate $\overline{w'T'}$.

The discrepancy between the car and the tripod for $\overline{w'^2}$ and $\overline{w'T'}$ may be related to a mismatch in the flux footprint or possibly related to the rapid flow distortion experienced at the location of the sonic anemometer on the vehicle. The road produces a distinct upward heat flux and an increase in $\overline{w'^2}$ on sunny days because it has a significantly lower albedo than the surrounding grasses and farmland. On 22 Aug we parked on the upwind side of the of the highway for approximately 30 min, but the car was also parked on the downwind side of the highway during assembly and disassembly of the tripod. For three

independent 8 min periods, the average $\overline{w'^2}$ and $\overline{w'T'}$ on the upwind side of the highway are measured at 0.15 $\mathrm{m^2 \, s^{-2}}$ and 0.085 $\mathrm{K \, m \, s^{-1}}$ respectively. Downwind of the highway $\overline{w'^2}$ and $\overline{w'T'}$ are found to be larger, near 0.33 $\mathrm{m^2 \, s^{-2}}$ and 0.109 $\mathrm{K \, m \, s^{-1}}$ on average (from 6 independent samples), which are more consistent with measurements made on the tripod. These findings for $\overline{w'^2}$ are similar to Gordon et al. (2012) who measured $\overline{w'^2}$ = 0.27 $\mathrm{m^2 \, s^{-2}}$ downwind of a four–lane highway on a sunny day.




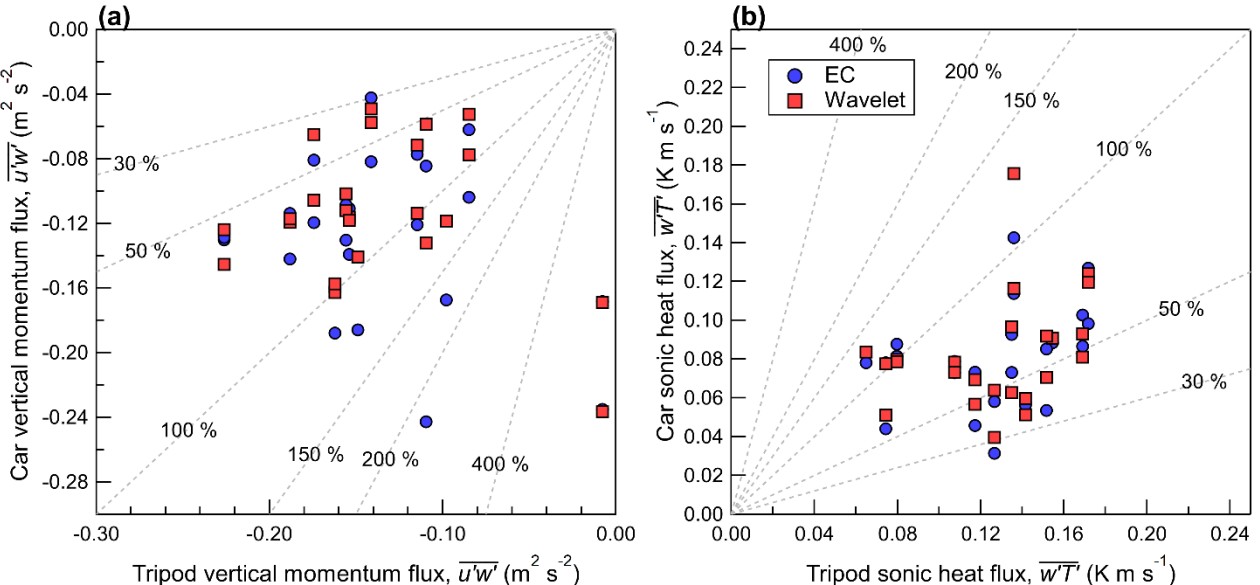

**Figure 8: The vertical momentum flux, $\overline{u'w'}$ (a) and the sonic heat flux, $\overline{w'T'}$ (b) measured by the tripod (horizontal) and compared to the mobile car (vertical). Covariances calculated using wavelet analysis and EC are shown as red and blue markers respectively. Dashed grey lines denote constant percentages of the independent variable.**

To investigate the flux footprint of the tripod versus the instrumented car, the footprint model of Kljun et al. (2015) is applied with $\bar{u} = 2.5$ m s$^{-1}$, a boundary layer height of $h = 1500$ m, a friction velocity of $u_* = 0.35$ m s$^{-1}$, an Obukhov length of $L = -30$ m, $\overline{v'^2} = 1.5$ m$^2$ s$^{-2}$ and a wind direction that is assumed perpendicular to the highway. These meteorological values represent estimations based on measurements made on 22 Aug. For the car $z_m \approx 1.7$ m but for the tripod $z_m \approx 1.4$ m. However, flow distortion on the mobile car results in the measurements being representative of a lower height than the height

at which the instrumentation is installed. Achberger and Bärring (1999) explored the displacement due to flow distortion on a mini–bus and estimated that the displacement at 2 m height was typically on the order of 0.2 m. Therefore, measurements obtained at $z_m = 1.7$ m on the mobile car in this study are probably representative of a slightly lower height between 1.5 and 1.6 m. For the upper height limit of $z_m = 1.7$ m, the footprint model predicts that the maximum location of influence to the flux is about 4.2 m upwind of the measurement location. For $z_m = 1.5$ m it is about 3.7 m upwind. Since the tripod is positioned

in the shoulder of the highway, 3.7 m upwind of the tripod is near the center of the highway. Assuming the instrumented car is in the lane closest to the tripod (or about 1.75 m from the edge of the highway), the maximum location of influence to the flux is near 6 m, or near the edge of the highway furthest from the tripod. Therefore, when the car is in the lane closest to the tripod, the measurements have a flux footprint that includes less influence from the highway. The footprint model predicts that the influence from the road is minimized when the car is driving in the lane furthest from the tripod, but for measurements

made during this study there is not a significant statistical difference in $\overline{w'^2}$ and $\overline{w'T'}$ for the close versus far highway lane.





The footprint model applied here is strongly impacted by the mean wind speed $\bar{u}$ – a lower $\bar{u}$ gives a location of maximum influence to the flux that is closer to the measurement system.

Another factor that may influence the velocity measurements made by the sonic anemometer is rapid distortion of the flow caused by the moving vehicle. Wyngard (1988) shows that the variance of scalar quantities (such as the sonic temperature
or a gas concentration) remains unchanged during rapid flow distortion. The velocity variances, however, may be altered during stretching and compression of the flow as it is forced to rise over the front end of the vehicle, in analogy to isotropic turbulence and flow over a symmetric hill (Britter et al., 1981; Gong and Ibbetson 1989). If it is assumed that the low bias in the measured $\overline{w'^2}$ on the car is caused by rapid flow distortion alone (i.e., no effect from the highway asphalt), then rapid distortion theory would predict a proportional increase in the velocity variance measured parallel to the vehicle motion.
However, in the case of the measurements of $\overline{w'^2}$ made during this study, there is likely a contribution from rapid distortion of the flow in addition to a contribution from the flux footprint mismatch between the car and tripod, but it is not possible to separate the effects in this work.

The vertical momentum fluxes, $\overline{u'w'}$ measured by the car and tripod are displayed in Fig. 8(a). For EC there is no significant bias for $\overline{u'w'}$ measured on the car compared to $\overline{u'w'}$ measured on the tripod. The tripod measurements of
$\overline{u'w'}$ generally fall within the 95% confidence interval of $\overline{u'w'}$ measured on the car (see Sect. 3.4.2). There are instances where $\overline{u'w'}$ measured by the two systems differ significantly however, and this suggests a better estimate of $\overline{u'w'}$ can probably be obtained by averaging multiple passes. The horizontal momentum flux, $\overline{u'v'}$, measured on the tripod does not agree with measurements made on the mobile car (not shown), and when sampling errors are considered $\overline{u'v'}$ measured on the car is not found to be significantly different than zero within the 66 % confidence interval.

**3.3 Velocity spectra**

Figure 9 displays the binned power spectral density (multiplied by frequency) of the velocity components for measurement Pass 5 (Fig. 9a), Pass 7 (Fig. 9b), and Pass 8 (Fig. 9c) from Track #1. These three measurement passes have been chosen since they demonstrate unique features in the car spectra, which are representative of the spectra from the remaining measurement passes not shown (see Fig. S6). The frequencies are normalized to give a wavelength as $\lambda = \bar{u}/f$ where $f$ is the frequency
(Hz) and $\bar{u}$ is the mean ambient wind on the tripod or the car relative flow on the mobile car. Each panel displays the spectra of $u$ (top), $v$ (middle) and $w$ (bottom). In general, the shape of the spectra measured on the mobile car agree well with the spectra measured by the tripod, however there are some notable differences: (1) Unlike the tripod, the power spectra of $u$ and $v$ measured on the car during Pass 7 and 8 increase at high frequencies ($\lambda < 5$ m). This increase may be related to white noise in the measured signal or perhaps aliasing and is present in about 75% of the measured spectra from Track #1. Langford et al.
(2015) show that the power spectra of the sonic temperature increase linearly with a +1 at high frequencies (in the inertial subrange) in the presence of white noise, resembling the findings in this study for $u$ and $v$. One potential source of white noise in the measured horizontal velocity components may be road unevenness (Schiehlen, 2006). Belusic et al. (2014) found distinct



peaks in their car measured $v$ spectra near a frequency of 7 Hz which they attribute to frame vibrations, and by comparing the sonic measurements to GPS–INS motion they concluded that road unevenness did not impact the high frequency portion of

the velocity spectra. (2) For $u$ in Pass 7 and 8 as $\lambda$ increases past 100 m, the power spectral density increases on the car while on the tripod the power spectral density decreases. (3) In Pass 7 and 8, $w$ appears to be under–sampled since the car spectra do not extend through the entire inertial subrange. Therefore, sampling at high vehicle speeds (> 15 m s$^{-1}$) would probably benefit from a sampling rate greater than 40 Hz. Additionally, in Pass 5 and 8 there is a general underestimation of the power spectral density of $w$ on the car compared to the tripod for $\lambda$ between about 5 to 80 m, and this underestimation is a common

feature in the measured car spectra.

**Figure 9: Binned power spectral density (multiplied by frequency) of the velocity components $u$ (top), $v$ (middle), and $w$ (bottom) measured by the roadside tripod (triangles) and the mobile car (circles). The frequencies are normalized to give a wavelength ($\lambda$) as**

**$\bar{u}/f$ where $f$ is the frequency (Hz) and $\bar{u}$ is the mean ambient wind speed (or car relative flow for the mobile car).**



### 3.4 Measurement uncertainties

#### 3.4.1 Flow distortion correction angle, $\theta$

Despite the rather strong relationship between the measured vertical velocity ($W$) and the measured longitudinal velocity ($U$) discussed in Sect. 2.4, there is still an uncertainty in the rotation angle ($\theta$) used to correct for the effect of flow distortion on the vertical velocity. The median of $\theta$ calculated using all binned values is 7.54°, with the lower and upper quartile (25th and 75th) being 7.38 and 7.70° respectively (IQR = 0.32°). If instead $\theta$ = Q25 = 7.38° is used for the flow distortion correction, the mean vertical velocity measured on the car during all measurement passes increases, giving $\bar{w}$ = 0.06 m s$^{-1}$ (using $\theta$ = Q50 = 7.54° gives $\bar{w}$ = 0.00 m s$^{-1}$). In addition, there is an increase in the magnitude of $\overline{w'^2}$, $\overline{w'T'}$ and $\overline{u'w'}$, giving a marginally better statistical agreement between the car and tripod for $\overline{w'^2}$ , $\overline{w'T'}$ as shown in Table 3. These results demonstrate that reducing $\theta$ to give $\bar{w} > 0$ m s$^{-1}$ is not sufficient to improve the agreement among all turbulence statistics and will not remove the bias noted in Sect. 3.2 for $\overline{w'^2}$ and $\overline{w'T'}$. Similarly, increasing $\theta$ from 7.54 ° does not remove the bias or improve the agreement between the car and tripod.

Table 3: Statistics calculated over all measurement passes (i.e., Track #1 and Track #2), but with $\theta$ = 7.38°.

|  | $\text{MBE}_{EC}$ | $\text{MBE}_W$ | $\text{RMSE}_{EC}$ | $\text{RMSE}_W$ | $\text{Mean}_{EC}$ Car | $\text{Mean}_W$ Car | $\text{Mean}_{EC}$ Tripod |
|---|---|---|---|---|---|---|---|
| $\overline{u'^2}$ | 0.89 | 0.44 | 1.43 | 0.75 | 2.14 | 1.68 | 1.26 |
| $\overline{v'^2}$ | 0.20 | 0.04 | 0.61 | 0.44 | 1.38 | 1.21 | 1.19 |
| $\overline{w'^2}$ | –0.10 | –0.11 | 0.11 | 0.12 | 0.18 | 0.17 | 0.29 |
| $\overline{u'w'}$ | –0.04 | –0.02 | 0.10 | 0.08 | –0.18 | –0.15 | –0.14 |
| $\overline{w'T'}$ | –0.04 | –0.03 | 0.05 | 0.05 | 0.09 | 0.09 | 0.13 |
| $\bar{u}$ | 0.04 | -- | 0.53 | -- | 2.44 | -- | 2.42 |

#### 3.4.2 Sampling errors

A significant concern when obtaining atmospheric measurements from an instrumented mobile car is the impact of sampling errors. Sampling errors on the mobile car may result from (i) the use of a record length that is too short to be representative of an ensemble mean, (ii) non–stationarity of the flow introduced by microscale variations or inhomogeneities in the terrain and surrounding structures (i.e., trees, buildings), or (iii) white noise and persistent structured signals introduced by vehicle resonance and vibrations.

In this work three methods to quantity the random measurement uncertainty are investigated: (1) the method of Finkelstein and Sims (2001) referred to as F&S (i.e., Eq. (17)), (2) the method of Mann and Lenschow (1994) referred to as M&L (i.e., Eq. (15)) and (3) the method of Lenschow et al. (2000) (Eq. (22)). F&S and M&L give an estimate of the overall random measurement uncertainty, while Lenschow et al. (2000) gives an estimate of random measurement uncertainty





attributed only to white noise in the measured signal. The method of Lenschow et al. (2000) does not include contributions from persistent structured signals that may occur at a specific frequency (i.e., from vehicle resonance or some other cause of vibrations such as speed bumps).

Figure 10 displays the random measurement uncertainty of the horizontal velocity variances ($\overline{u'^2}$ and $\overline{v'^2}$) measured

on the car plotted as a function of the magnitude of the variance. Likewise, Fig. 11 shows the random measurement uncertainty of the vertical velocity variance ($\overline{w'^2}$) and Fig. 12 displays the random measurement uncertainty of the measured covariances ($\overline{u'w'}$ and $\overline{w'T'}$). The random uncertainty estimates calculated from M&L and F&S agree well on the mobile car platform for velocity variances when $m = 30$ s. However, for $m = 30$ s F&S tends to give a slightly greater magnitude of random measurement uncertainty than M&L for covariances (i.e., Fig. 12). This is similar to the findings of Finkelstein and Sims

(2002), who note that method of F&S contains a contribution from both the autocovariance and cross–covariance function leading to a larger magnitude and more conservative estimate of the sampling error compared to M&L. Rannik et al. (2016) note that F&S gives an estimate of the "total" random measurement uncertainty.

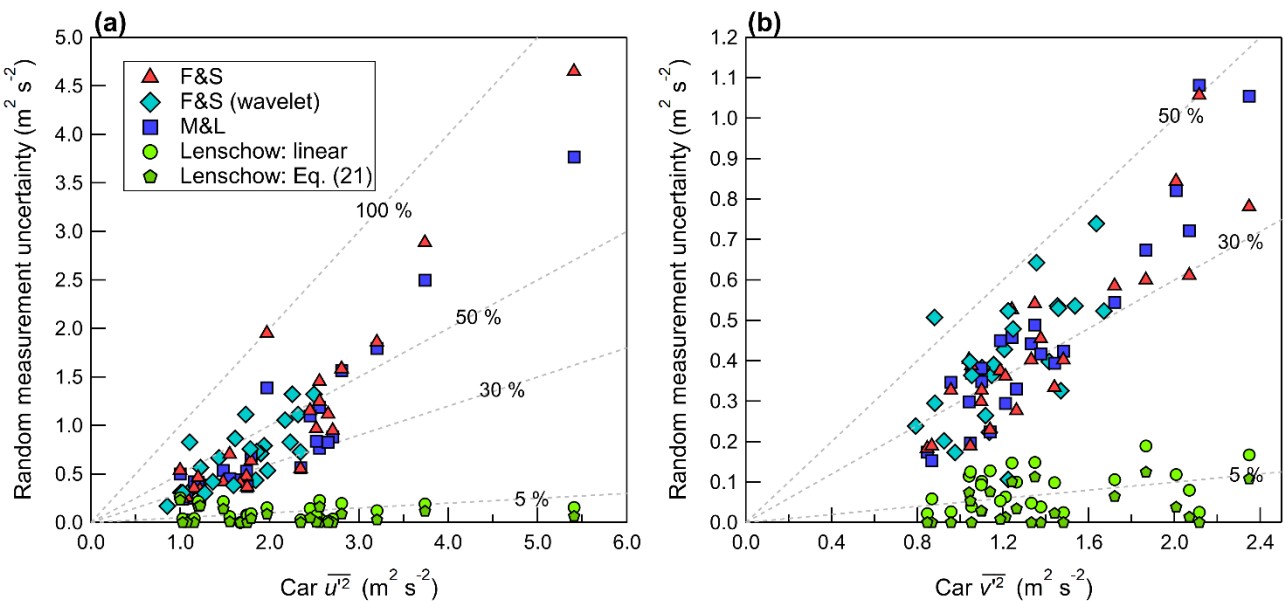

**Figure 10: Random measurement uncertainty of the horizontal velocity variance measured on the car, plotted as a function of (a) the longitudinal velocity variance $\overline{u'^2}$ and (b) the lateral velocity variance, $\overline{v'^2}$. Dashed grey lines denote constant percentages of the independent variable.**



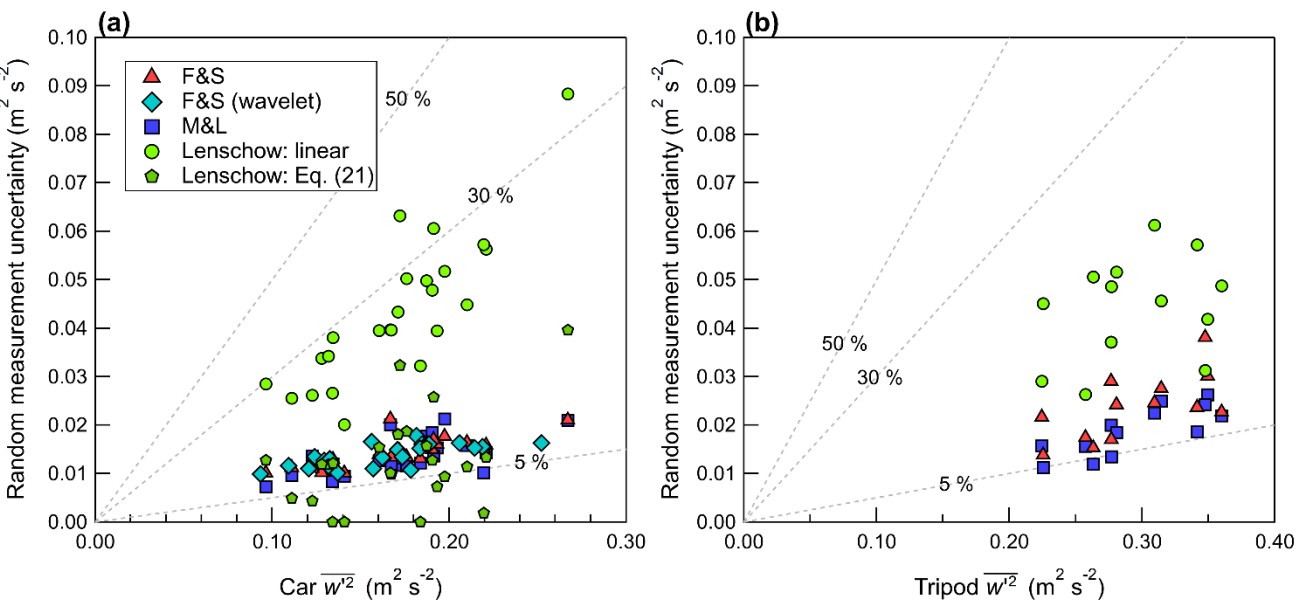

**Figure 11: Random measurement uncertainty of the vertical velocity variance measured (a) on the car and (b) on the tripod, plotted as a function of $\overline{w'^2}$. Dashed grey lines denote constant percentages of the independent variable.**

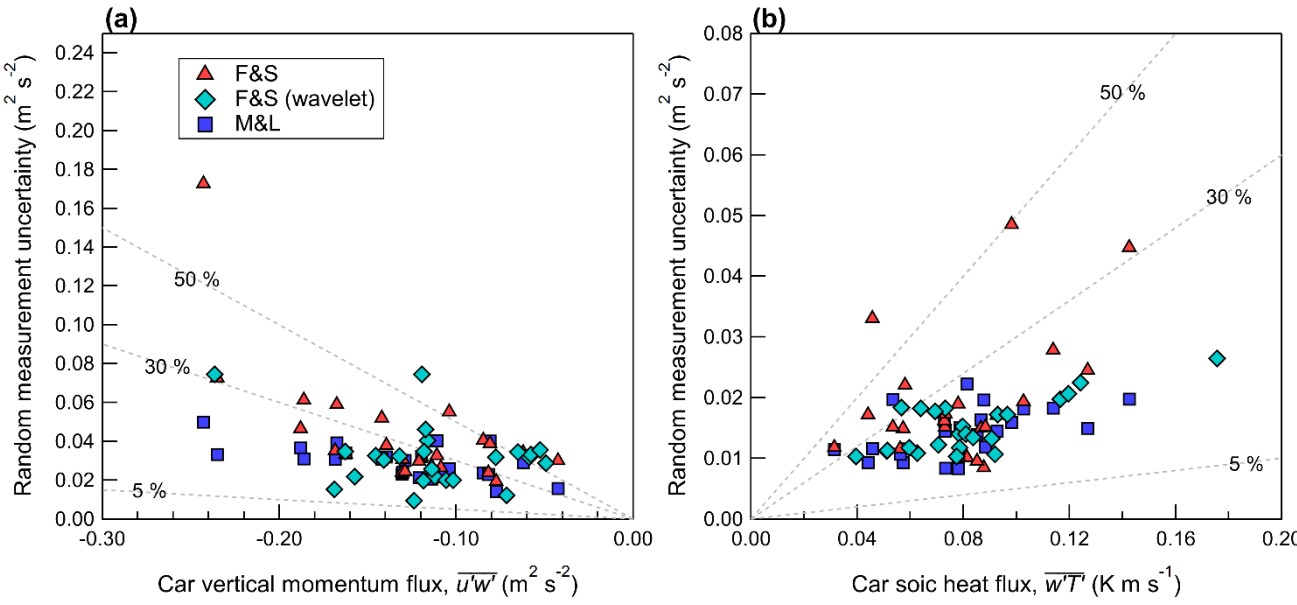

**Figure 12: Measurement uncertainty of (a) the vertical turbulent momentum flux $\overline{u'w'}$ and (b) the vertical turbulent sonic heat flux $\overline{w'T'}$ measured on the car and plotted as a function of the flux magnitude. Dashed grey lines denote constant percentages of the independent variable.**





The random measurement uncertainty calculated from F&S and M&L scales approximately linearly with increasing magnitude of the velocity variance or covariance, as shown in Fig. 10 to Fig. 12. For Track #2 there are several instances where

$\overline{u'^2}$ is large (i.e., 2 to 5 m² s⁻²) and $\delta_{FS}$ is on the order of $\overline{u'^2}$. $\delta_{FS}$ and $\delta_{ML}$ give 1 standard deviation of the random measurement uncertainty of a measured variance or covariance (Rannik et al., 2016). Thus, $\overline{u'^2}$ measured on Track #2 is not significantly different than 0 in the 95 % confidence interval for some measurement passes. A trend in the velocity record results in an autocorrelation function that does not fall to zero as expected and instead remains elevated at large time lags. This suggests that $\delta_{FS}$ in this study includes a contribution from non–stationarity in the record, which is consistent with the conclusions of

Rannik et al. (2016) for measurements made on stationary towers, who found that $\delta_{FS}$ continues to increase as $m$ is increased to 300 s.

Reconstructing the time series using wavelet analysis produces a filtered time series, where the resolved low frequency contributions are excluded. Applying F&S to the reconstructed time series gives an estimate of $\delta_{FS}$ for the wavelet variances and covariances (shown in Figs. 10 to 12 as diamonds). For $\overline{u'^2}$, wavelet estimates of $\delta_{FS}$ follow a similar trend to

the uncertainty estimates found using the unfiltered time series, that is, as the magnitude of the wavelet variance increases, so does $\delta_{FS}$. However, for times when wavelet analysis predicts a smaller $\overline{u'^2}$, $\delta_{FS}$ is also found to be proportionally reduced, and $\overline{u'^2}$ on most passes becomes consistent with the tripod in the 95% confidence interval.

Figures 10 and 11 show the random measurement uncertainty due to white noise in the measured signal ($\delta_L$) estimated according to Lenschow et al. (2000). For the measurement tracks investigated here the use of a linear fit to estimate $\delta_L$ gives

a much larger uncertainty than Eq. (21). In the case of the vertical velocity, $\delta_L$ estimated using a linear fit extrapolation is 3 to 4 times larger than the total random measurement uncertainty according to $\delta_{FS}$. $\delta_L$ is expected to represent a contribution to the total random measurement uncertainty and therefore $\delta_L < \delta_{FS}$ (Rannik et al., 2016). This suggests that the linear fit significantly underestimates the true variance and overestimates the amount of white noise for $w$. If a power law fit (Eq. (21)) is used instead of a linear fit, $\delta_L$ is reduced and for several measurements passes $\delta_L < \delta_{FS}$. The difficulty estimating $\delta_L$ for $w$

on the car is not unexpected, since $w$ has an integral time scale (ITS) of 0.05 to 0.1 s for vehicle speeds near 20 m s⁻¹ and this is only 2 to 4 times the sampling interval of the sonic anemometer. This limits the amount of autocovariance function time lags that lie within the inertial subrange, giving a poor fit. Lenschow et al. (2000) note that for a successful power law fit to the autocovariance function, the ITS must be "several times larger" than the sampling interval of the instrument. For $w$ measured on the tripod, the use of Eq. (21) gives undefined $\overline{\epsilon'^2}$ while a linear fit gives $\delta_L > \delta_{FS}$ as shown in Fig. 11b.

Compared to $w$, the measured horizontal velocity components on the car ($u$, $v$) have a larger ITS (on the order of 1 s) and a larger signal–to–noise ratio (SNR). Rannik et al. (2016) argue that the method proposed by Lenschow et al. (2000) is best suited for closed–path sensors as opposed to open–path sensors and high–precision instrumentation such as sonic anemometers. They found that the method of Lenschow et al. (2000) gives a relatively unbiased estimate of the white noise when the SNR is small and applied the method to estimate $\delta_L$ only for $w$ (not for $u$ or $v$). For $u$ and $v$ in this study, $\delta_L$ typically

represents a small contribution to the total random measurement uncertainty, except for weaker signals (i.e., lower measured





horizontal variances). The presence of white noise in the measured $u$ and $v$ signals is also supported by the spectra shown in Fig. 9b, where a near +1 slope appears at high frequencies within the inertial subrange. This is not the case for $w$, where the spectra do not show a +1 slope at high frequencies and hence $w$ spectra have no evidence of white noise impacting the measured signal. This may suggest that $\delta_L$ overestimates the magnitude of white noise present in $w$, and so $\delta_L$ is likely not a

reliable estimate of white noise in the vertical velocity for car measurements made at high vehicle speeds near 20 m s$^{-1}$.

In addition to Track #1 and Track #2, the car was driven on a gravel road at relatively high vehicle speeds ($s$ between 20 and 23 m s$^{-1}$) for a short (< 5 min) period. The effect of the gravel road is investigated by splitting the short period into non–overlapping intervals of 49 s (yielding 5 unique samples), and performing the same analysis as outlined in Sect. 2. The car measurements on the gravel road are similar to car measurements obtained on the paved road, for a comparable $s$. The

magnitude of the variances and covariances on the gravel road are consistent with those measured on the paved road within the 95 % confidence interval, and the uncertainty estimates ($\delta_{FS}$, $\delta_{ML}$ and $\delta_L$) are the same order of magnitude. The measured velocity variances and uncertainty analysis for the gravel road are displayed in the supplemental material (Fig. S7). These measurements suggest that the road surface types investigated in this study have a limited influence on the measured turbulence statistics.

**3.4.3 Tripod velocity record contamination from passing traffic**

Since the study was designed to investigate measurements in non–idealized conditions, the highway locations have public access and therefore other vehicle traffic was present during the measurements. The traffic consisted largely of passenger vehicles (such as cars, pickup trucks, sport utility vehicles and minivans), but the traffic on 22 Aug was more significant and was comprised of occasional large trucks (dump trucks and tractor–trailers). For measurement passes on 22 Aug (with video

recordings available on the tripod), the dashcam recorded between 26 and 40 total passing vehicles, of which 0 to 4 were large trucks. The car takes about 45 s to complete a track, but on the tripod the equivalent averaging period is between 6 to 8 min. For some measurement passes, the mobile car does not experience any traffic contamination, but this is not the case for the tripod. Therefore, the tripod will measure a different composition and amount of passing traffic than the car, potentially leading to differences in the measurements made by the two systems.

Large trucks produce a significant amount of vehicle–induced turbulence, but passenger cars and sport utility vehicles produce much less in comparison (Miller et al., 2019; Gordon et al., 2012). Furthermore, the wake has limited lateral spread relative to the vehicle travel direction (Kim et al., 2018), except perhaps for times with significant advection, so the most noticeable effect on the tripod will be from traffic in the adjacent highway lane (i.e., closet to the tripod). For measurements on the car, passing traffic (particularly large trucks) is found to enhance the measured velocity variances (i.e., Fig. 7c). Like

the car, the main effect of passing traffic on the tripod measurements would also be an enhancement of the velocity variances. Thus, for times when there is no traffic contamination on the car, the differences shown in Fig. 6 between the car and tripod– measured velocity variances may be underestimated, since the tripod velocity variances are enhanced due to passing traffic,





but the car measurements are not. Therefore, the presence of traffic measured by the tripod and not the car introduces an additional uncertainty into the measurement comparisons shown in Sect. 3.

**4 Conclusions**


The results presented in Sect. 3 demonstrate that the instrumented car design used in this study can successfully measure the mean atmospheric boundary layer close to the surface, but the car measurements may vary significantly based on the surrounding features such as trees, buildings, and other traffic. Therefore, the interpretation of the car–based measurements depends largely on the specific application, since the car may measure turbulence that is localized and not represented in

single–point measurements made at a stationary tower. In the previous study of Belušic et al. (2014) there was limited upwind surface obstructions and no other traffic during their measurements. Despite the more idealized environment, their measurements revealed times when the horizontal velocity variances ($\overline{u'^2}$ and $\overline{v'^2}$) measured on the car were significantly larger than a nearby stationary tower, and they suggest that intense, temporally limited flow structures are to blame. These events dominate the measurements made on the car, but not on the tripod since the averaging period is longer. In this

investigation, $\overline{u'^2}$ and $\overline{v'^2}$ on the car calculated using EC are also found to be much larger than measured on tripod for some measurement passes (i.e., a factor between 2 and 5 for $\overline{u'^2}$ with $\mathrm{RMSE}_{EC\ Car}/\mathrm{Mean}_{EC\ Tripod} = \mathrm{NRMSE}_{EC} \approx 114\%$). When the measurement uncertainty in Sect 3 is considered, these large $\overline{u'^2}$ are not significantly different than 0 in the 95% confidence interval, since $\delta_{FS} \approx \overline{u'^2}$. Applying wavelet analysis to calculate $\overline{u'^2}$ and $\overline{v'^2}$ gives significantly reduced magnitudes for some measurement passes, particularly those measurement passes with the largest estimated EC variances. This results in an

improved agreement between the mobile car and tripod for $\overline{u'^2}$ and $\overline{v'^2}$ (for $\overline{u'^2}$ $\mathrm{RMSE}_{W\ Car}/\mathrm{Mean}_{EC\ Tripod} = \mathrm{NRMSE}_{W} \approx$ 60%). The improved agreement using wavelet analysis suggests that wavelet analysis resolves length scales near and exceeding the length of the measurement track (i.e., 1000 m); in this study the change in surface features on Track #2 (from a windbreak to an open field) may yield an artificial low frequency contribution in the velocity record. Thus, when measuring from an instrumented car it is important to be aware of changes in terrain and land usage, which can strongly impact the near–ground

measurements.

Evidence from this investigation shows that passing traffic (especially large trucks) can also lead to an increase in the velocity variances measured on the car. However, if the passing traffic is sporadic, the resulting increase in the measured velocity variances from vehicle–induced turbulence can be identified and removed using wavelet analysis. For a measurement pass in this study that experienced a passing heavy–duty truck and sport utility vehicle, removing the times when the traffic

passes the mobile car (9 out of 46 s) decreases the turbulent kinetic energy by about 10%. This highlights the importance of video recordings in conjunction with sonic anemometer measurements on a car, so that times with possible traffic contamination can be identified in applications where its measurement is not intended.

The sampling uncertainties in Sect. 3 suggest that it is possible to measure a statistically significant vertical momentum flux on the mobile car at vehicle speeds near 20 m s⁻¹. $\overline{u'w'}$ measured on the car is typically found to consistent with the tripod



within the 95% confidence interval, but for some passes $\overline{u'w'}$ measured on the car is small ($< 0.06$ m$^2$ s$^{-2}$) and not significantly

different than 0 in the 95% confidence interval. Therefore, for measurements obtained on the mobile car a better estimate of

$\overline{u'w'}$ can probably be obtained by averaging multiple passes with a spatial extent of 10's of kilometers. Random measurement

uncertainty estimates of $\overline{u'w'}$ by F&S and M&L (which give 1 standard deviation of the uncertainty) have magnitudes that are

typically 10 to 40% of the measured flux. Furthermore, there is no significant bias in $\overline{u'w'}$ measured on the car when the entire

set of measurement passes is considered ($\text{MBE}_{EC\ Car}/\text{Mean}_{EC\ Tripod} = \text{NMBE}_{EC} \approx -4\%$ and $\text{MBE}_{W\ Car}/\text{Mean}_{EC\ Tripod} =$

$\text{NMBE}_W \approx -14\%$).

The vertical velocity ($\overline{w'^2}$) and vertical sonic heat flux ($\overline{w'T'}$) measured in this study are found to be biased low compared

to measurements made on the tripod ($\text{NMBE}_{EC} \approx -38\%$ for both $\overline{w'^2}$ and $\overline{w'T'}$). The low bias on the car is probably due to the

combination of two factors: (1) the footprint measured by the car contains less of the low–albedo highway than the tripod and

(2) rapid flow distortion at the measurement location on the car. Interestingly, there is evidence of a similar low bias in $\overline{w'^2}$

(but not $\overline{w'T'}$) measured by the car in Belusic et al. (2014), where only 4 out of the 19 completed passes measured a greater

$\overline{w'^2}$ on the car than the stationary tower (i.e., their Fig. 4). This demonstrates that wind tunnel testing or computational flow

modelling of each specific instrumented car design may be useful to quantify the effects of rapid flow distortion on the

measured velocity variances and covariances. Applying the method of Lenschow et al. (2000) to estimate the magnitude of

white noise in the measured vertical velocity signal at vehicle speeds near 20 m s$^{-1}$ likely underestimates the true signal variance

and overestimates the amount of white noise and therefore is not recommended.

The mean wind speed and mean wind direction were found to be consistent with measurements made on the tripod. For

$\bar{u}$ measured on Track #1 and Track #2, the NMBE $\approx 6\%$ and NRMSE $\approx 0\%$ respectively. Even a short averaging period of 10 s

for car measurements made at a vehicle speed near 20 m s$^{-1}$ provides a reliable estimate of mean wind direction on the car; for

about 250 unique intervals on 20 and 22 Aug the interquartile range of the wind direction is 42 and 34° respectively. Despite

the rather consistent wind direction, the mean wind speed in any individual 10 s averaging period may vary considerably; the

interquartile range for $\bar{u}$ is 1.3 and 1.9 m s$^{-1}$ on 20 and 22 Aug respectively. The large variation in the 10 s mean wind speed

likely represents more localized flow that exists in a specific location. Therefore, the instrumented car may prove invaluable

for studies that require precise measurement of localized flow, providing simultaneous measurement of wind speed and

direction over a large domain. This study shows that even when the sonic anemometer is placed particularly close to the vehicle

(compared with Belusic et al. (2014), for example), it is still possible to correct for flow distortion effects and obtain

measurements of the mean wind and turbulence that are consistent (within the 95% confidence interval) of those measured by

a nearby stationary tripod.

The results presented in this investigation demonstrate that car–based measurements of turbulence require care when

selecting the appropriate spatial and temporal averaging, and when selecting the measurement location, to ensure that the

measurements obtained are representative of the specific application. This is demonstrated in our measurements, where the



highway surface/flux footprint, upwind obstructions and passing traffic are all found to have a significant effect on the measured values but are not necessarily errors since they do represent real features that can generate atmospheric turbulence.

**Code and data availability**

The data used to generate the figures and complete the analysis presented herein is available online at https://doi.org/10.5683/SP3/IBBDTF.

**Author contribution**

Stefan Miller and Mark Gordon designed and performed the experiment. Stefan Miller completed the analysis of the measured data and complied the results. Stefan Miller prepared the manuscript with contributions from Mark Gordon.

**Competing interests**

The authors declare that they have no conflict of interest.

**Acknowledgements**

We thank Peter Taylor for providing the SUV used in this study and for his assistance during the experimental data collection (as a driver in the experiment) on 30 Aug. This research was supported by National Sciences and Engineering Research Council
of Canada – NSERC (grant no. RGPIN 2015–04292).

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
