# Peer review of "The measurement of mean wind, variances and covariances from an instrumented mobile car in a rural environment"

_Atmospheric Measurement Techniques, 2022_

## Author Comment (AC1)

We thank referee #1 for taking the time to read our manuscript and provide useful suggestions and feedback. We have modified the manuscript to address your points. Referee comments are in black, our **responses are in green**, and **changes to the manuscript are colored blue**. Our line references refer to the revised manuscript.

**Referee #1**

The paper presents a thorough evaluation of mobile car-mounted turbulence measurements near the surface. The mobile measurements are compared with corresponding stationary tower data, which shows that the mobile system can provide satisfactory mean and turbulence data following a proper procedure for flow distortion correction. Furthermore, it is shown that using wavelet analysis for calculating higher order statistics of the mobile measurements can be more appropriate than the traditional eddy-covariance technique. The paper is well written and I recommend publication after minor review.

**Specific comments:**

**1. It is not clear how many measurement passes are made for each track.**

**Response:** We have added new details to the revised manuscript to address this point.

**Added lines 151 - 161:** Track #1 and Track #2 overlap spatially for 380 m, and so a portion of the data contained within both measurement tracks are identical, for each trip past the tripod. Table 1 gives the number of measurement passes performed on each measurement track. The amount of measurement passes that are excluded (from both Track #1 and Track #2) due to traffic ahead of the instrumented car is also given. Two extra measurement passes corresponding only to Track #2 were also analyzed on 22 Aug, where the car was parked at the tripod and then drove away (a constant vehicle speed was achieved before 120 m). Since the car did not travel down the entire length of Track #1 prior to parking at the roadside, there are no corresponding Track #1 for these two measurement passes on Track #2.

**Table 1**: The number of measurement passes performed on each measurement track on 20 and 22 Aug.

| Date | Track 1 | Track 2 | Excluded (traffic ahead) | No. of trips past the tripod |
|------|---------|---------|--------------------------|------------------------------|
| 20 Aug | 6 | 6 | 1 | 7 |
| 22 Aug | 5* | 7* | 2 | 9 |

* Two extra measurement passes are included corresponding only to Track #2, where the car was stationary prior to the pass. There is no corresponding Track #1 since the car did not complete the entire length of Track #1 before parking near the tripod.

**2. In 243: The authors should clarify how exactly the mobile data can have "a time series with a temporal length 11 times that of" the 1–km variance. Since the track length is 1 km, where does the additional data (the temporal equivalent of 10 km) come from?**

**Response:** The wavelet coefficients are calculated following the software developed by Torrence and Compo (1988), which applies the convolution theorem, and hence makes use of the Fourier transform. The Fourier transform assumes the data is periodic, and this periodicity causes errors at the start and end of the wavelet transform calculated from a finite measurement record of temporal length $T$, known as edge effects (Torrence and Compo, 1988). These edge effects occur because stretched wavelets (i.e., representing long time scales) can extend beyond the boundaries of the measurement period, and into regions where no data exists. Therefore, if we apply wavelet analysis to a time series of only length $T$ (i.e., the same $T$ that eddy-covariance is applied to), there will be some information at large wavelet scales (particularly near the start and end of the time series) that is not reliable. This results in an unreliable wavelet variance or covariance for the time scales of interest (i.e., for wavelet scales $a^* \leq T$) when calculated over $T$. In an attempt to rectify this problem, Torrence and Compo (1988) recommend padding the time series of length $T$ with zeros (but this too has drawbacks). However, in this study there is no need to pad the time series before or after $T$ with artificial data (or zeros), since the instrumented car continued driving down the same road after measuring on Track #1 and Track #2, providing continuous measured data before and after each measurement pass. This continuous data limits edge effects in our wavelet variances and covariances calculated over $T$ (for the time scales of interest), providing a more reliable estimate for each measurement pass. Hence, when performing wavelet analysis, we include additional data before and after the 1000 m track (equivalent to a spatial distance of about 10 km) that comes from continuous driving in the vicinity of the tripod at a relatively constant speed, and in most cases on the same road. The instrumented car did not come to rest, except briefly at a stop sign or to reverse direction. There are two exceptions for measurement passes on Track #2, where the car initially started from rest and reached a constant speed before travelling 120 m from the tripod.

Torrence and Compo (1988) define the cone of influence as "the maximum period of useful information at that particular time" which is determined as an e-folding time "chosen so that wavelet power for a discontinuity at the edge drops by a factor $e^{-2}$". In our study wavelet analysis is performed on a time series of length $11T$, with the measurement pass located between $5 \leq T < 6$. Based on the cone of influence definition by Torrence and Compo (1988), wavelet coefficients for each measurement pass are primarily influenced by data between $3.63 \leq T < 7.37$ for $a^* \leq T$. Therefore, the data between $0 \leq T < 3.63$ and $7.37 \leq T < 11$ have little impact on the calculated wavelet variance or covariance, and thus are not necessary to give a reliable estimate for the measurement pass.

**Edited and expanded lines 256 – 274:** In Eq. (10) index value $a^*$ represents the maximum (Fourier equivalent) wavelet scale and controls the time scales that are included in the wavelet variance, which in this work is set to match $T_m$ as closely as possible. $G_n^x(a_j)$ is calculated from a measured time series with a temporal length of $11T_m$, where the data corresponding to the measurement pass (over which $\sigma_{x\,1\,\mathrm{km}}^2$ is calculated) are located at the center of this period (i.e., from $5 \leq T_m < 6$). This approach is applied to ensure that the wavelet transform coefficients used to calculate the wavelet variances are not impacted by edge effects for scales up to $a^*$ (i.e., they do not lie outside of the cone of influence), while still retaining good computational efficiency (Torrence and Compo, 1988; Schaller et al., 2017). Torrence and Compo (1988) recommend zero padding a finite series of length $T_m$ to reduce edge effects, but in this study, there is no need to pad the time series before or after the measurement pass with zeros, since the instrumented car continued driving down the same road after measuring on Track #1 and Track #2, providing continuous measured data before and after each measurement pass. This continuous data limits edge effects in the wavelet variances and covariances calculated over $5 \leq T_m < 6$ for $a^* \leq T_m$, providing a more reliable estimate for each measurement pass. Hence, the additional data before and after the measurement pass (equivalent to a spatial distance of about 10 km) comes from continuous driving in the vicinity of the tripod at a relatively constant speed, and in most cases on the same road. The instrumented car did not come to rest, except briefly at a stop sign or to reverse direction. There are two exceptions for measurement passes on Track #2, where the car initially started from rest and reached a constant speed before travelling 120 m from the tripod. Based on the cone of influence definition by Torrence and Compo (1988), wavelet coefficients for each measurement pass are primarily influenced by data between $3.63 \leq T_m < 7.37$ for $a^* \leq T_m$. Therefore, the data between $0 \leq T_m < 3.63$ and $7.37 \leq T_m < 11$ have little impact on the

calculated wavelet variance or covariance, and thus are not necessary to give a reliable estimate for the measurement pass.

**3. lns 605–607 and 617: The interpretation of the confidence interval should be clarified. Why is one standard deviation related to the 95% confidence interval?**

**Response:** We have expanded lines 647 – 654 to clarify the definition of confidence interval (see below).

**Why is the confidence related to "not significantly different than 0" at ln 607 and "consistent with the tripod" at ln 617?**

**Response**: For some measurement passes the 95% confidence interval of $\overline{u'^2}_{EC\,car}$ includes zero, suggesting these variances are not statistically different than zero in the 95 % confidence interval. When the confidence interval of a variance or covariance measured on the car includes the value measured on the tripod, then measurements are deemed consistent between the two systems in that confidence interval for that measurement pass. We have corrected line 617.

**Edited and moved to beginning of section (line 647-654):** $\delta_{FS}$ and $\delta_{ML}$ give 1 standard deviation of the random measurement uncertainty of a measured variance or covariance for the averaging period $T$, which Rannik et al. (2009) demonstrate is nearly equivalent to the standard error of the variance or covariance. Thus, in this work we define the 68 % confidence interval as the range $F \pm \delta$, and likewise the 95 % confidence interval as the range $F \pm 1.96\delta$, where $F$ is the measured variance or covariance. When the confidence interval of a variance or covariance includes the value measured on the tripod, then measurements are deemed consistent between the two systems in that confidence interval (for that measurement pass).

**Edited line 677-678:** However, for times when wavelet analysis predicts a smaller $\overline{u'^2}$, $\delta_{FS}$ is also found to be proportionally reduced.

**Line 20,591,676,749,765:** Changed 'significantly' to 'statistically'.

**Technical comments:**

**– In 365: the sentence looks unfinished?**

**Response:** The sentence has been updated.

**Edited line 423 - 424:** The mean wind speed shown in Fig. 3.5 (b) shows relatively good agreement between the car and tripod with no significant bias ($\mathrm{MBE}_{car}/\bar{u}_{tripod}$ = 2 % and $\mathrm{RMSE}_{car}/\bar{u}_{tripod}$ = 22 %).

**– In 394: "Figure 6(a), (b) and (c) show..." – Such statements should be clear from figure captions and are not needed in the main text. Similar holds for other figures (e.g. Fig. 8).**

**Response:** We have modified the revised manuscript to remove such statements.

**Deleted**: Figure 11 displays the random measurement uncertainty of the horizontal velocity variances ($\overline{u'^2}$ and $\overline{v'^2}$) measured on the car plotted as a function of the magnitude of the variance. Likewise, Fig. 12 shows the random measurement uncertainty of the vertical velocity variance ($\overline{w'^2}$) and Fig. 13 displays the random measurement uncertainty of the measured covariances ($\overline{u'w'}$ and $\overline{w'T'}$).

**Replaced with:** Figures 11 to 13 display the random measurement uncertainty of the measured variances and covariances, calculated using these three methodologies.

**– In 478: delete one occurrence of "of the".**

**Response:** This has been corrected in the revised manuscript.

**– Fig. 12b: x–axis should say "... sonic...".**

**Response:** This has been corrected in the revised manuscript.

**Other minor corrections/changes:**

1. Changed $T$ to $T_m$ to represent the averaging period in the updated manuscript. In the original manuscript $T$ is also being used for sonic temperature, which may lead to confusion.

2. **Corrected Line 784-785 in conclusion section**: For $\bar{u}$ measured on Track #1 and Track #2, the NMBE ≈ 2 % and NRMSE ≈ 22 % respectively.

3. Other grammar fixes (i.e., missing "the" and "or").

**References**

Rannik, Ü., Mammarella, I., Aalto, P., Keronen, P., Vesala, T., & Kulmala, M. (2009). Long-term aerosol particle flux observations part I: Uncertainties and time-average statistics. Atmospheric Environment, 43(21), 3431–3439. https://doi.org/10.1016/j.atmosenv.2009.02.049

Torrence, C., & Compo, G. P.: A practical guide to wavelet analysis, Bulletin of the American Meteorological Society, 79(1), 61–78, https://doi.org/10.1175/1520-0477(998)079<0061:apgtwa>2.0.co;2, 1998.

---

## Author Comment (AC2)

**We thank referee #2 for taking the time to read our manuscript and provide useful suggestions and feedback. We have modified the manuscript to address your points. Referee comments are in black, our responses are in green, and changes to the manuscript are colored blue. Our line references refer to the revised manuscript.**

**Referee #2**
**General comments.**

In this work the authors compare turbulence measurements made on a car instrumented with a sonic anemometer with the same measurements taken on a fixed tripod at the side of the road. The choice of the site, with lateral obstructions but with light traffic, is appropriate to the purposes of the comparison. The growing need of spatially extended data over inhomogeneous terrain makes mobile measurements an important topic in turbulence measurements. The most intriguing part is a wavelet based approach to reduce eddy-covariance measurements when they substantially differ from the ones from the fixed instruments and to remove the effect of intersecting vehicles on the measurements. The paper is clear and well written, sometimes a bit heavy to read for someone not familiar with all the correction methods described throughout. I recommend publication after the authors addressed my minor comments.

**Major comments.**

**1) Section 2.5. As said above, this may be the most interesting part of the paper, since it offers a solid correction method for car measurements. However, while the wavelet analytical formulation is very clear, how the wavelet is applied is far less clear. I struggled a bit in understanding what is the averaging time scale on which the measurements are compared. At line 403 it seems that the maximum track length (in seconds) is around 40–60s while 5 to 8 minutes averaging was used before for wind directions and speeds. Was a different averaging time used for variances and covariances to compare with wavelet analysis or was only T set to 40–60 s as wavelet max–scale to reduce low–frequency contribution?**

**Response:** We have expanded Section 2.8 to address your question and clarify the averaging periods used (see below).

**Expanded Section 2.8 between lines 371 to 402:**

The averaging period ($T_m$) on the car is set to the temporal length of the 1000 m track for atmospheric means. For car-measured atmospheric variances and covariances $T_m$ is calculated from Taylor's hypothesis (as $T_m = L/\bar{u}$,) with an $L$ = 1000 m track length. On the instrumented car we have $\bar{u} \cong s$, where $s$ is the near-constant vehicle speed over the 1000 m track, and therefore $T_m$ is equivalent to the time it takes for the car to travel 1000 m (for both eddy-covariance and wavelet analysis). For the car, any measurement pass that follows closely behind a vehicle is excluded from the results. To quantify a wavelet variance or covariance on the car, the maximum wavelet time scale ($a^*$) must be chosen. In this study $a^*$ is set to match $T_m$ as closely as possible (i.e., the temporal length of the 1000 m track). This approach is used so that the wavelet variance (or covariance) is directly comparable to eddy covariance since both methodologies will include the same time scales ($a^*$ controls the maximum time scale included in the wavelet variance or covariance).

$T_m$ on the tripod is set to 5 min for atmospheric means, but for atmospheric variances and covariances $T_m$ varies depending on the mean 5-min wind speed measured by the tripod ($\bar{u}$) according to Taylor's frozen hypothesis, where $L$ = 1000 m. For the two measurement days investigated here, $T_m$ on the tripod ranges between 5 and 8 min. For consistency, the averaging period used for calculation of the tripod means, variances and covariances is centered on the time that the instrumented car passes the tripod (for both Track #1 and Track #2). The choice of $L$ on the tripod is not trivial, since $L$ should be determined by taking into consideration the vehicle speed in addition to the mean ambient flow. Since the mean ambient flow in this study was relatively weak (~2.5 m s$^{-1}$) and typically at an angle to the vehicle, we have $\bar{u} \cong s$ on the car, but in strong ambient flow $\bar{u} \neq s$, and Taylor's hypothesis would suggest a different $L$ on the tripod to compare with the 1000 m track driven by the car. For example, if $\bar{u}$ = 30 m s$^{-1}$ on the car with $s$ = 22 m s$^{-1}$, then 1000 m travelled by the car would correspond to a distance of $L_s$ = 1364 m travelled by an air parcel, and this distance should be used to determine $T_m$ on the tripod, that is $T_m = \frac{L_s}{\bar{u}} > \frac{1000}{\bar{u}}$. The averaging periods adopted in this study for each methodology (wavelet analysis or eddy-covariance) and measurement system (car or tripod) are summarized in Table 2.

**Table 2:** The averaging periods ($T_m$) used to calculate means, variances and covariances on the instrumented car and stationary roadside tripod. $T_m$ for variances and covariances are calculated from Taylor's hypothesis ($T_m = L/\bar{u}$) with an $L$ = 1000 m track length. On the instrumented car we have $\bar{u} \cong s$, where $s$ is the near-constant vehicle speed over the 1000 m track, but for the tripod $\bar{u}$ is the mean wind speed measured on the tripod and calculated from 5-min averages.

| | $T_m$ on the instrumented car | $T_m$ on the tripod |
|---|---|---|
| **Means** | 40 – 60 s | 5 min |
| **Variances / covariances (eddy covariance)** | 40 – 60 s | Varies between 5 and 8 min |
| **Variances / covariances (wavelet analysis)** | 40 – 60 s, including wavelet scales up to $a^*$, where $a^* \cong T_m$ | N/A |

**2) The comparison between the turbulent heat flux is very interesting and well discussed. Would not be the case to compare the temperature variances seen by the tripod and the car?**

**Response:** We have added analysis of the sonic temperature variance to the revised manuscript.

**New figure for sonic temperature variance (line 552 – 555):**

[Figure]

**Figure 9: The sonic temperature variance, $\overline{T'^2}$ measured by the tripod (horizontal) and compared to the mobile car (vertical). Covariances calculated using wavelet analysis and eddy covariance are shown as red and blue markers respectively. Dashed grey lines denote constant percentages of the independent variable.**

**Added new line for temperature variance to Table 3:**

Table 3: Statistics calculated over all measurement passes (i.e., on both tracks on 20 and 22 Aug). Subscript $EC$ denotes a statistical variance or a covariance calculated using eddy–covariance. A subscript $W$ denotes a variance or covariance calculated using wavelet analysis.

| | $MBE_{EC}$ | $MBE_W$ | $RMSE_{EC}$ | $RMSE_W$ | $Mean_{EC}$ Car | $Mean_W$ Car | $Mean_{EC}$ Tripod |
|---|---|---|---|---|---|---|---|
| $\overline{u'^2}$ (m² s⁻²) | 0.90 | 0.44 | 1.44 | 0.75 | 2.15 | 1.69 | 1.26 |
| $\overline{v'^2}$ (m² s⁻²) | 0.20 | 0.04 | 0.61 | 0.44 | 1.38 | 1.21 | 1.19 |
| $\overline{w'^2}$ (m² s⁻²) | −0.11 | −0.12 | 0.12 | 0.13 | 0.17 | 0.16 | 0.29 |
| $\overline{T'^2}$ (K²) | **0.05** | **0.01** | **0.19** | **0.18** | **0.52** | **0.48** | **0.46** |
| $\overline{u'w'}$ (m² s⁻²) | 0.005 | 0.02 | 0.08 | 0.08 | −0.13 | −0.11 | −0.14 |
| $\overline{w'T'}$ (K m s⁻¹) | −0.05 | −0.04 | 0.06 | 0.06 | 0.08 | 0.08 | 0.13 |
| $\overline{u}$ (m s⁻¹) | 0.04 | | 0.53 | | 2.45 | | 2.42 |

**Added lines 527 – 533**: Despite a low bias noted in $\overline{w'T'}$, there is no low bias found in the sonic temperature variance ($\overline{T'^2}$) measured on the instrumented car compared to the tripod (shown in Fig. 9), where the $MBE_{EC} = 0.05$ K². Since the sonic anemometer is placed over the front bumper which holds the vehicle engine, there may potentially be some impact from its heat in our measurements. While the effect of engine heat is probably more important in cold ambient temperatures, there may still be an impact on $T$ measured on the car in this study while driving, which would likely result in $\overline{T'^2}$ being biased high compared to an instrumented car without engine heat effects.

**Expanded lines 537 – 544:** For three independent 8 min periods, the average $\overline{w'^2}$, $\overline{T'^2}$ and $\overline{w'T'}$ on the upwind side of the highway are measured at 0.15 m² s⁻², 0.46 K² and 0.085 K m s⁻¹ respectively. Downwind of the highway $\overline{w'^2}$, $\overline{T'^2}$ and $\overline{w'T'}$ are found to be larger, near 0.33 m² s⁻², 0.68 K² and 0.109 K m s⁻¹ on average (from 5 independent samples), which are more consistent with measurements made on the tripod, except for $\overline{T'^2}$. The car-measured $\overline{T'^2}$ on the downwind side of the highway has a large standard deviation (0.41 K²) and a single outlier that skews the average. Removing this outlier (where $\overline{T'^2}$ = 1.39 K²) reduces the average car-measured $\overline{T'^2}$ downwind of the highway to 0.50 K², which is more consistent with the tripod; the 8 min sample with the anomalously large $\overline{T'^2}$ does not have an anomalously large $\overline{w'^2}$ or $\overline{w'T'}$.

**Minor comments.**

**3) The paper would benefit a table with the number of measurements records analyzed each day.**

**Response:** We have added new details to the revised manuscript to address this point.

**Added lines 151 - 161:** Track #1 and Track #2 overlap spatially for 380 m, and so a portion of the data contained within both measurement tracks are identical, for each trip past the tripod. Table 1 gives the number of measurement passes performed on each measurement track. The amount of measurement passes that are excluded (from both Track #1 and Track #2) due to traffic ahead of the instrumented car is also given. Two extra measurement passes corresponding only to Track #2 were also analyzed on 22 Aug, where the car was parked at the tripod and then drove away (a constant vehicle speed was achieved before 120 m). Since the car did not travel down the entire length of Track #1 prior to parking at the roadside, there are no corresponding Track #1 for these two measurement passes on Track #2.

**Table 1**: The number of measurement passes performed on each measurement track on 20 and 22 Aug.

| Date | Track 1 | Track 2 | Excluded (traffic ahead) | No. of trips past the tripod |
|---|---|---|---|---|
| **20 Aug** | 6 | 6 | 1 | 7 |
| **22 Aug** | 5* | 7* | 2 | 9 |

\* Two extra measurement passes are included corresponding only to Track #2, where the car was stationary prior to the pass. There is no corresponding Track #1, since the car did not complete the entire length of Track #1 before parking near the tripod.

**4) In Section 2.4 what is the averaging time of the data presented?**

**Response:** We have added new details to the manuscript to address this question.

**Added lines 201 – 206:** The data shown in Fig. 3 includes all back–and–forth passes completed on 20 and 22 Aug and the binned data are derived from individual measurements made by the 40 Hz sonic anemometer (every 0.025 s). Each bin requires at least 80 independent samples (2 s of data), otherwise it is rejected. Binning using individual measurements is done instead of averaging over all of A and over all of B, since it is difficult to maintain a constant vehicle speed during each part of the measurement pass. However, most measurements of $U$ fall into 2 to 4 speed bins during a particular back-and-forth pass consisting of parts A and B.

**Other minor corrections/changes:**

1. Changed $T$ to $T_m$ to represent the averaging period in the updated manuscript. In the original manuscript $T$ is also being used for sonic temperature, which may lead to confusion.

2. **Corrected Line 784-785 in conclusion section**: For $\bar{u}$ measured on Track #1 and Track #2, the $\mathrm{NMBE} \approx 2\ \%$ and $\mathrm{NRMSE} \approx 22\ \%$ respectively

3. Other grammar fixes (missing "the", "or").

---

## Author Response (AR1)

We thank the two anonymous referees for their time and valuable feedback; likewise, we thank the editor for their time. We have modified the manuscript to address the points raised by the referees and we feel the manuscript has been strengthen as a result of their feedback. Aside from expanded text in the revised manuscript to address the specific comments by each referees, we have included new analysis related to the sonic temperature variance, as suggested by Referee #2.

**Referee comments are in bolded black, our** responses are in green, and changes to the manuscript are colored blue. Our line references to follow refer to the tracked changes manuscript, which is appended to the end of this file.

**Referee #1 (RC1)**

The paper presents a thorough evaluation of mobile car-mounted turbulence measurements near the surface. The mobile measurements are compared with corresponding stationary tower data, which shows that the mobile system can provide satisfactory mean and turbulence data following a proper procedure for flow distortion correction. Furthermore, it is shown that using wavelet analysis for calculating higher order statistics of the mobile measurements can be more appropriate than the traditional eddy-covariance technique. The paper is well written and I recommend publication after minor review.

**Specific comments:**

1. It is not clear how many measurement passes are made for each track.

**Response:** This is valid point. We have added new details to the revised manuscript to address this point.

Added lines 151 - 161: Track #1 and Track #2 overlap spatially for 380 m, and so a portion of the data contained within both measurement tracks are identical, for each trip past the tripod. Table 1 gives the number of measurement passes performed on each measurement track. The amount of measurement passes that are excluded (from both Track #1 and Track #2) due to traffic ahead of the instrumented car is also given. Two extra measurement passes corresponding only to Track #2 were also analyzed on 22 Aug, where the car was parked at the tripod and then drove away (a constant vehicle speed was achieved before 120 m). Since the car did not travel down the entire length of Track #1 prior to parking at the roadside, there are no corresponding Track #1 for these two measurement passes on Track #2.

| Date   | Track 1 | Track 2 | Excluded (traffic ahead) | No. of trips past the tripod           7 |  |  |
|--------|---------|---------|--------------------------|------------------------------------------|--|--|
| 20 Aug | 6       | 6       | 1                        |                                          |  |  |
| 22 Aug | 5*      | 7*      | 2                        | 9                                        |  |  |

| Table | 1: | The 1 | number | of | measurement | passes | performed | on each | measurement | track | on 2 | 20 | and | 22 A | ug. |
|-------|----|-------|--------|----|-------------|--------|-----------|---------|-------------|-------|------|----|-----|------|-----|
|-------|----|-------|--------|----|-------------|--------|-----------|---------|-------------|-------|------|----|-----|------|-----|

\* Two extra measurement passes are included corresponding only to Track #2, where the car was stationary prior to the pass. There is no corresponding Track #1 since the car did not complete the entire length of Track #1 before parking near the tripod.

**2. In 243: The authors should clarify how exactly the mobile data can have "a time series with a temporal length 11 times that of" the 1–km variance. Since the track length is 1 km, where does the additional data (the temporal equivalent of 10 km) come from?**

**Response:** The wavelet coefficients are calculated following the software developed by Torrence and Compo (1988), which applies the convolution theorem, and hence makes use of the Fourier transform. The Fourier transform assumes the data is periodic, and this periodicity causes errors at the start and end of the wavelet transform calculated from a finite measurement record of temporal length T, known as edge effects (Torrence and Compo, 1988). These edge effects occur because stretched wavelets (i.e., representing long time scales) can extend beyond the boundaries of the measurement period, and into regions where no data exists. Therefore, if we apply wavelet analysis to a time series of only length T (i.e., the same T that eddycovariance is applied to), there will be some information at large wavelet scales (particularly near the start and end of the time series) that is not reliable. This results in an unreliable wavelet variance or covariance for the time scales of interest (i.e., for wavelet scales  $a^* \leq T$ ) when calculated over T. In an attempt to rectify this problem, Torrence and Compo (1988) recommend padding the time series of length T with zeros (but this too has drawbacks). However, in this study there is no need to pad the time series before or after T with artificial data (or zeros), since the instrumented car continued driving down the same road after measuring on Track #1 and Track #2, providing continuous measured data before and after each measurement pass. This continuous data limits edge effects in our wavelet variances and covariances calculated over T (for the time scales of interest), providing a more reliable estimate for each measurement pass. Hence, when performing wavelet analysis, we include additional data before and after the 1000 m track (equivalent to a spatial distance of about 10 km) that comes from continuous driving in the vicinity of the tripod at a relatively constant speed, and in most cases on the same road. The instrumented car did not come to rest, except briefly at a stop sign or to reverse direction. There are two exceptions for measurement passes on Track #2, where the car initially started from rest and reached a constant speed before travelling 120 m from the tripod.

Torrence and Compo (1988) define the cone of influence as "the maximum period of useful information at that particular time" which is determined as an e-folding time "chosen so that wavelet power for a discontinuity at the edge drops by a factor  $e^{-2}$ ". In our study wavelet analysis is performed on a time series of length 11*T*, with the measurement pass located between  $5 \le T < 6$ . Based on the cone of influence definition by Torrence and Compo (1988), wavelet coefficients for each measurement pass are primarily influenced by data between  $3.63 \le T < 7.37$  for  $a^* \le T$ . Therefore, the data between  $0 \le T < 3.63$  and  $7.37 \le T < 11$  have little impact on the calculated wavelet variance or covariance, and thus are not necessary to give a reliable estimate for the measurement pass.

Edited and expanded lines 256 – 274: In Eq. (10) index value  $a^*$  represents the maximum (Fourier equivalent) wavelet scale and controls the time scales that are included in the wavelet variance, which in this work is set to match  $T_m$  as closely as possible.  $G_n^x(a_j)$  is calculated from a measured time series with a temporal length of  $11T_m$ , where the data corresponding to the measurement pass (over which  $\sigma_{x_1 \text{ km}}^2$  is calculated) are located at the center of this period (i.e., from  $5 \le T_m < 6$ ). This approach is applied to ensure that the wavelet transform coefficients used to calculate the wavelet variances are not impacted by edge effects for scales up to  $a^*$  (i.e., they do not lie outside of the cone of influence), while still retaining good computational efficiency (Torrence and Compo, 1988; Schaller et al., 2017). Torrence and Compo (1988) recommend zero padding a finite series of length  $T_m$  to reduce edge effects, but in this study, there

is no need to pad the time series before or after the measurement pass with zeros, since the instrumented car continued driving down the same road after measuring on Track #1 and Track #2, providing continuous measured data before and after each measurement pass. This continuous data limits edge effects in the wavelet variances and covariances calculated over  $5 \le T_m < 6$  for  $a^* \le T_m$ , providing a more reliable estimate for each measurement pass. Hence, the additional data before and after each measurement pass (equivalent to a spatial distance of about 10 km) comes from continuous driving in the vicinity of the tripod at a relatively constant speed, and in most cases on the same road. The instrumented car did not come to rest, except briefly at a stop sign or to reverse direction. There are two exceptions for measurement passes on Track #2, where the car initially started from rest and reached a constant speed before travelling 120 m from the tripod. Based on the cone of influence definition by Torrence and Compo (1988), wavelet coefficients for each measurement pass are primarily influenced by data between  $3.63 \le T_m < 7.37$  for  $a^* \le T_m$ . Therefore, the data between  $0 \le T_m < 3.63$  and  $7.37 \le T_m < 11$  have little impact on the calculated wavelet variance or covariance, and thus are not necessary to give a reliable estimate for the measurement pass.

**3.** Ins 605–607 and 617: The interpretation of the confidence interval should be clarified. Why is one standard deviation related to the 95% confidence interval?

**Response:** We have expanded lines 654 – 661 to clarify the definition of confidence interval (see below).

Why is the confidence related to "not significantly different than 0" at ln 607 and "consistent with the tripod" at ln 617?

**Response**: For some measurement passes the 95% confidence interval of  $\overline{u'^2}_{EC\ car}$  includes zero, suggesting these variances are not statistically different than zero in the 95% confidence interval. When the confidence interval of a variance or covariance measured on the car includes the value measured on the tripod, then measurements are deemed consistent between the two systems in that confidence interval for that measurement pass. We have corrected line 617, now lines 700-701.

Edited line 689-691 and moved to beginning of section (line 654-661):  $\delta_{FS}$  and  $\delta_{ML}$  give 1 standard deviation of the random measurement uncertainty of a measured variance or covariance for the averaging period  $T_m$ , which Rannik et al. (2009) demonstrate is nearly equivalent to the standard error of the variance or covariance. Thus, in this work we define the 68 % confidence interval as the range  $F \pm \delta$ , and likewise the 95 % confidence interval as the range  $F \pm 1.96\delta$ , where F is the measured variance or covariance. When the confidence interval of a variance or covariance includes the value measured on the tripod, then measurements are deemed consistent between the two systems in that confidence interval (for that measurement pass).

Edited line 700-701: However, for times when wavelet analysis predicts a smaller  $\overline{u'^2}$ ,  $\delta_{FS}$  is also found to be proportionally reduced., and  $\overline{u'^2}$  on most passes becomes consistent with the tripod in the 95% confidence interval.

Line 21,600,691,767,786: Changed 'significantly' to 'statistically'.

**Technical comments:**

- In 365: the sentence looks unfinished?

Response: The sentence has been updated, thank you for pointing this out.

**Edited line 426 - 427:** The mean wind speed shown in Fig. 3.5 (b) shows relatively good agreement between the car and tripod with no significant bias (MBEcar/ $\bar{u}_{tripod} = 2$  % and RMSEcar/ $\bar{u}_{tripod} = 22$  %).

- In 394: "Figure 6(a), (b) and (c) show..." – Such statements should be clear from figure captions and are not needed in the main text. Similar holds for other figures (e.g. Fig. 8).
Response: We have modified the revised manuscript to remove such statements, as listed below:

1) Deleted from line 457: Figure 6(a), (b) and (c) sho  $\overline{u'^2}$ ,  $\overline{v'^2}$  and  $\overline{w'^2}$  respectively.

2) Deleted from line 530: Figure 6(c) displays  $\overline{w'^2}$  measured on the mobile car compared to  $\overline{w'^2}$  measured on the tripod.

3) Deleted 534-535: Figure 8(a) displays the vertical momentum flux  $(\overline{u'w'})$  and Fig. 8(b) shows the sonic heat flux  $(\overline{w'T'})$ . Figure 8 follows the same conventions as Fig. 6. Inserted on Line 535: "(shown in Fig. 8(b)"

4) Deleted line 594: The vertical momentum fluxes,  $\overline{u'w'}$  measured by the car and tripod are displayed in Fig. 8(a). Inserted on line 595: "as shown in Fig. 8(a)"

5) Deleted 662-665: Figure 10 displays the random measurement uncertainty of the horizontal velocity variances  $(\overline{u'^2} \text{ and } \overline{v'^2})$  measured on the car plotted as a function of the magnitude of the variance. Likewise, Fig. 11 shows the random measurement uncertainty of the vertical velocity variance  $(\overline{w'^2})$  and Fig. 12 displays the random measurement uncertainty of the measured covariances  $(\overline{u'w'} \text{ and } \overline{w'T'})$ . Replaced with on line 659-661: Figures 11 to 13 display the random measurement uncertainty of the measurement uncertainty

6) Deleted from line 702-703: Figures 10 and 11 show the random measurement uncertainty due to white noise in the measured signal ( $\delta_L$ ) estimated according to Lenschow et al. (2000). Inserted on line 704: "as shown in Figs. 11 and 12".

**\*Note figure references of inserted text have been updated to account for the additional figure added in the revised manuscript for sonic temperature variance (Fig. 9).**

– In 478: delete one occurrence of ''of the''.Response: This has been corrected in the revised manuscript on line 546 of tracked changes manuscript.

Fig. 12b: x-axis should say "... sonic...".Response: This has been corrected in the revised manuscript (line 682).

**Referee #2 (RC2)**

**General comments.**

In this work the authors compare turbulence measurements made on a car instrumented with a sonic anemometer with the same measurements taken on a fixed tripod at the side of the road. The choice of the site, with lateral obstructions but with light traffic, is appropriate to the purposes of the comparison. The growing need of spatially extended data over inhomogeneous terrain makes mobile measurements an important topic in turbulence measurements. The most intriguing part is a wavelet based approach to reduce eddy-covariance measurements when they substantially differ from the ones from the fixed instruments and to remove the effect of intersecting vehicles on the measurements. The paper is clear and well written, sometimes a bit heavy to read for someone not familiar with all the correction methods described throughout. I recommend publication after the authors addressed my minor comments.

**Major comments.**

1) Section 2.5. As said above, this may be the most interesting part of the paper, since it offers a solid correction method for car measurements. However, while the wavelet analytical formulation is very clear, how the wavelet is applied is far less clear. I struggled a bit in understanding what is the averaging time scale on which the measurements are compared. At line 403 it seems that the maximum track length (in seconds) is around 40–60s while 5 to 8 minutes averaging was used before for wind directions and speeds. Was a different averaging time used for variances and covariances to compare with wavelet analysis or was only T set to 40–60 s as wavelet max–scale to reduce low–frequency contribution?

**Response:** We have expanded Section 2.8 to address your specific question and clarify the averaging periods used (see below).

**Expanded Section 2.8 between lines 376 to 402:**

The averaging period  $(T_m)$  on the car is set to the temporal length of the 1000 m track for atmospheric means. For car-measured atmospheric variances and covariances  $T_m$  is calculated from Taylor's hypothesis (as  $T_m = L/\bar{u}$ ,) with an L = 1000 m track length. On the instrumented car we have  $\bar{u} \cong$ *s*, where *s* is the near-constant vehicle speed over the 1000 m track, and therefore  $T_m$  is equivalent to the time it takes for the car to travel 1000 m (for both eddy-covariance and wavelet analysis). For the car, any measurement pass that follows closely behind a vehicle is excluded from the results. To quantify a wavelet variance or covariance on the car, the maximum wavelet time scale ( $a^*$ ) must be chosen. In this study  $a^*$  is set to match  $T_m$  as closely as possible (i.e., the temporal length of the 1000 m track). This approach is used so that the wavelet variance (or covariance) is directly comparable to eddy covariance since both methodologies will include the same time scales ( $a^*$  controls the maximum time scale included in the wavelet variance or covariance).

 $T_m$  on the tripod is set to 5 min for atmospheric means, but for atmospheric variances and covariances  $T_m$  varies depending on the mean 5-min wind speed measured by the tripod ( $\bar{u}$ ) according to Taylor's frozen hypothesis, where L = 1000 m. For the two measurement days investigated here,  $T_m$  on the tripod ranges between 5 and 8 min. For consistency, the averaging period used for calculation of the tripod means, variances and covariances is centered on the time that the instrumented car passes the tripod (for both Track #1 and Track #2). The choice of L on the tripod is not trivial, since L should be determined by

taking into consideration the vehicle speed in addition to the mean ambient flow. Since the mean ambient flow in this study was relatively weak (~2.5 m s-1) and typically at an angle to the vehicle, we have  $\bar{u} \cong s$  on the car, but in strong ambient flow  $\bar{u} \neq s$ , and Taylor's hypothesis would suggest a different *L* on the tripod to compare with the 1000 m track driven by the car. For example, if  $\bar{u} = 30$  m s-1 on the car with s = 22 m s-1, then 1000 m trackel driven by the car would correspond to a distance of  $L_s = 1364$  m travelled by an air parcel, and this distance should be used to determine  $T_m$  on the tripod, that is  $T_m = \frac{L_s}{\bar{u}} > \frac{1000}{\bar{u}}$ . The averaging periods adopted in this study for each methodology (wavelet analysis or eddy–covariance) and measurement system (car or tripod) are summarized in Table 2.

Table 2: The averaging periods  $(T_m)$  used to calculate atmospheric means, variances and covariances on the instrumented car and stationary roadside tripod.  $T_m$  for variances and covariances are calculated from Taylor's hypothesis  $(T_m = L/\bar{u})$  with an L = 1000 m track length. On the instrumented car we have  $\bar{u} \cong s$ , where s is the near-constant vehicle speed over the 1000 m track, but for the tripod  $\bar{u}$  is the mean wind speed measured on the tripod and calculated from 5-min averages.

|                                               | $T_m$ on the instrumented car                                              | $T_m$ on the tripod        |  |  |
|-----------------------------------------------|----------------------------------------------------------------------------|----------------------------|--|--|
| Means                                         | 40 - 60  s                                                                 | 5 min                      |  |  |
| Variances / covariances
(eddy covariance)  | 40 – 60 s                                                                  | Varies between 5 and 8 min |  |  |
| Variances / covariances
(wavelet analysis) | $40-60$ s, including wavelet scales up to $a^*$ ,
where $a^* \cong T_m$ | N/A                        |  |  |

2) The comparison between the turbulent heat flux is very interesting and well discussed. Would not be the case to compare the temperature variances seen by the tripod and the car?

**Response:** This is an excellent suggestion. We have added analysis of the sonic temperature variance to the revised manuscript.

[revised manuscript text omitted]

**4) In Section 2.4 what is the averaging time of the data presented?**

**Response:** We have added new details to the manuscript to address this question.

Added lines 202-206: The data shown in Fig. 3 includes all back–and–forth passes completed on 20 and 22 Aug and the binned data are derived from individual measurements made by the 40 Hz sonic anemometer (every 0.025 s). Each bin requires at least 80 independent samples (2 s of data), otherwise it is rejected. Binning using individual measurements is done instead of averaging over all of A and over all of B, since it is difficult to maintain a constant vehicle speed during each part of the measurement pass. However, most measurements of U fall into 2 to 4 speed bins during a particular back–and–forth pass consisting of parts A and B.

**Other minor corrections/changes:**

- 1. Changed T to  $T_m$  throughout to represent the averaging period in the revised manuscript. In the original manuscript T is also being used for sonic temperature which may lead to confusion.
- 2. Updated figure and table numbers throughout to account for the new tables (Table 1 and 2) and the new figure (Fig. 9) added to address Referee comments.
- 3. Lines 920-922: Inserted reference to Rannik et al. (2009).
- 4. Corrected Line 803 (conclusion section): "For  $\bar{u}$  measured on Track #1 and Track #2, the NMBE  $\approx$  2 % and NRMSE  $\approx$  22 % respectively."
- 5. Added units to Table 5 (previous Table 3).
- 6. Updated all '-' to '-' throughout, except in references.
- 7. Grammar fixes:
  - a. Line 243: changed "scaled-averaged" to "scale-averaged".
  - b. Line 352: inserted "is".
  - c. Line 358: inserted "to".
  - d. Line 511 (caption of Fig. 7): inserted "of".

**References**

[revised manuscript text omitted]

---

## Author Response (AR2)

**We again would like to thank the editor for their time, and both referees for their time and feedback.**

**You have uploaded a new supplement pdf\*.file. If you applied any changes to the supplement PDF, a tracked changes file is mandatory. Please include the tracked-changes of your manuscript together with the tracked-changes of your supplement in one PDF and upload them as the "Author's tracked changes".**

**Response:** There have been no changes to the supplement PDF since the initial submission. We apologize for the confusion.

**Minor correction in label of Figure 13(b)**

We have corrected units of the sonic heat flux in Figure 13(b) from "$m^2 \, s^{-2}$" to "$K \, m \, s^{-1}$" (y-axis).